# G-protein-coupled receptor diversity and evolution in the closest living relatives of metazoa

Alain Garcia De Las Bayonas[1,2]*, Nicole King[1,2]

[1]Department of Molecular and Cell Biology, University of California, Berkeley, Berkeley, United States; [2]Howard Hughes Medical Institute, United States, Berkeley, United States

## eLife Assessment

This **important** study fills a gap in our knowledge of the evolution of GPCRs in holozoans, as well as the phylogeny of associated signaling pathway components such as G proteins, GRKs, and RIC8 proteins. The evidence supporting the conclusions is **compelling**, with the analysis of extensive new genomic data from choanoflagellates and other non-animal holozoans. Overall, the study is thorough and well-executed. It will be a resource for researchers interested in both the comparative genomics of multicellularity and GPCR biology more broadly, especially given the importance of GPCRs as highly druggable targets

**\*For correspondence:**
alaingdlb@berkeley.edu

**Competing interest:** The authors declare that no competing interests exist.

**Abstract** G-protein-coupled receptors (GPCRs) play a pivotal role in the perception of environmental cues across eukaryotic diversity. Although GPCRs have been relatively well characterized in metazoans, GPCR signaling is poorly understood in their sister group, the choanoflagellates, and in other close relatives of metazoans (CRMs). Here, we examine GPCR diversity and evolution in choanoflagellates by curating a catalog of 918 GPCRs, 141 G proteins, and 367 associated regulators from 23 choanoflagellate genomes and transcriptomes. We found that the repertoire of choanoflagellate GPCRs is larger and more diverse than previously anticipated, with 18 GPCR families found in choanoflagellates, of which 12 families are newly identified in these organisms. Comparative analyses revealed that most choanoflagellate GPCR families are conserved in metazoans and/or other eukaryotic lineages. Adhesion GPCRs and a class of GPCRs fused to kinases (the GPCR-TKL/Ks) are the most abundant GPCRs in choanoflagellates. The identification of GPCR repertoires in CRMs and other non-metazoans refines our understanding of metazoan GPCR evolution and reveals the existence of previously unreported GPCR families in metazoans and at the root of the eukaryotic tree.

## Introduction

G-protein-coupled receptors (GPCRs) constitute one of the largest and oldest families of receptors used to sense extracellular cues in eukaryotes (*Nordstrom et al., 2011*; *Krishnan et al., 2012*; *de Mendoza et al., 2014*). A conserved seven-transmembrane (7TM) domain is a hallmark of GPCRs, while the wide spectrum of extracellular and intracellular domains in some GPCRs reflects the diversification of the gene family and its functions (*Lagerström and Schiöth, 2008*). For example, the extracellular N-terminus and the three extracellular loops of the 7TM domain respond to a wide range of cues, including odorant molecules, peptides, amines, lipids, nucleotides, and other molecules (*Yang et al., 2021*). Ligand binding triggers a conformational change in the intracellular loops, thereby activating heterotrimeric GTP-binding proteins (G proteins) and additional regulators to shape

downstream cellular responses (*Latorraca et al., 2017*). Thus, GPCRs are essential for the control of metazoan development and tissue homeostasis, and their dysregulation often leads to pathological conditions in humans, making them the most researched drug targets in the pharmaceutical industry (*Hauser et al., 2017*).

Based on pioneering phylogenetic analyses in vertebrates, most metazoan GPCRs have been classified into five major families: Glutamate, Rhodopsin, Adhesion, Frizzled, and Secretin (GRAFS; *Fredriksson et al., 2003*; *Schiöth and Fredriksson, 2005*; *Lagerström and Schiöth, 2008*). Later studies found that four of these families (Glutamate, Rhodopsin, Adhesion, and Frizzled), alongside cyclic AMP (cAMP) receptors, GPR-107/108-like GPCRs, GPCR PIPKs, and GPR180, exist in most metazoans and in diverse other eukaryotic lineages (*Bakthavatsalam et al., 2006*; *Kamesh et al., 2008*; *Nordström et al., 2008*; *Krishnan et al., 2012*; *Krishnan et al., 2014*; *de Mendoza et al., 2014*; *van den Hoogen et al., 2018*; *Mojib and Kubanek, 2020*; *Hall et al., 2023*; *Luo et al., 2023*). Nonetheless, the premetazoan ancestry of many metazoan GPCRs remains unclear.

With the increasing availability of genomic and transcriptomic datasets from diverse early branching metazoans (*Srivastava et al., 2010*; *Guzman and Conaco, 2016*; *Gold et al., 2019*; *Nong et al., 2020*; *Schultz et al., 2021*; *Francis et al., 2023*; *Santini et al., 2023*; *Steffen et al., 2023*; *Vargas et al., 2024*), the closest living relatives of metazoans, the choanoflagellates (*King et al., 2008*; *Fairclough et al., 2013*; *Richter et al., 2018*; *Brunet et al., 2019*; *Hake et al., 2024*), and other holozoans, including members of Filasterea, Ichthyosporea, and Corallochytrea (*Suga et al., 2013*; *Torruella et al., 2015*; *Grau-Bové et al., 2017*; *Hehenberger et al., 2017*; *Ocaña-Pallarès et al., 2022*; *Sarre et al., 2024*; *Figure 1A*), we set out to refine our understanding of GPCR evolution.

To this end, we analyzed the GPCR repertoires of 23 choanoflagellate species and diverse other eukaryotes. Because of the structural modularity of GPCRs, the canonical 7TM domain on one side and the associated N-terminal and C-terminal regions on the other can follow distinct evolutionary trajectories. In addition, many receptors grouped under the same GPCR family (e.g. Rhodopsins and aGPCRs) are subdivided into numerous subfamilies. Therefore, we described and compared GPCRomes at three different levels in our analysis: (1) family- and (2) subfamily-level, based on the 7TM sequences of diverse GPCRs, and (3) associated protein domain composition, based on their respective N- and C-terminal regions.

## Results

### Identification of GPCRs and downstream signaling pathway components in the proteomes of 23 diverse choanoflagellates

To catalog the diversity of GPCRs and components of the downstream signaling pathway encoded by choanoflagellates, we analyzed the genome-derived and transcriptome-derived predicted proteomes of 23 choanoflagellate species (Supplementary file 1; *King et al., 2008*; *Fairclough et al., 2013*; *Richter et al., 2018*; *Brunet et al., 2019*; *Hake et al., 2024*). We first surveyed for candidate choano-flagellate GPCRs by searching choanoflagellate proteomes with Hidden Markov Models (HMMs) from the 7TM domains of 54 previously identified GPCR families (https://www.ebi.ac.uk/interpro/set/pfam/CL0192/; *Figure 1—figure supplement 1(1)*; Supplementary file 2). In a complementary approach, we used an HMM that reflects the topology of the 7TM domains of all GPCRs (GPCRHMM; *Wistrand et al., 2006*). These two approaches recovered 1095 and 1070 putative choanoflagellate GPCRs, respectively, of which 381 sequences were shared between the two datasets, leaving 1784 unique GPCR candidates. To remove potential false positives, highly fragmented sequences, and isoforms, the GPCR candidates were then subjected to additional filtering through homology- and topology-based approaches; 1113 sequences were removed after these filtering steps (*Figure 1—figure supplement 1(2)*).

Because existing 7TM HMMs may be biased toward metazoan sequences, we used the recovered choanoflagellate GPCRs to generate new choanoflagellate-derived 7TM HMMs (Supplementary file 3). To this end, the 671 validated choanoflagellate GPCRs were sorted by sequence similarity, resulting in 18 clusters (*Table 1*, Materials and methods 'Recovering additional choanoflagellate GPCRs using choanoflagellate GPCR BLAST queries and custom choanoflagellate GPCR HMMs' and 'Clustering of the 918 validated choanoflagellate GPCRs'); the 76 GPCRs that did not assort into any of these clusters were kept for downstream analyses. We then built 7TM HMMs for each of the 18 choanoflagellate

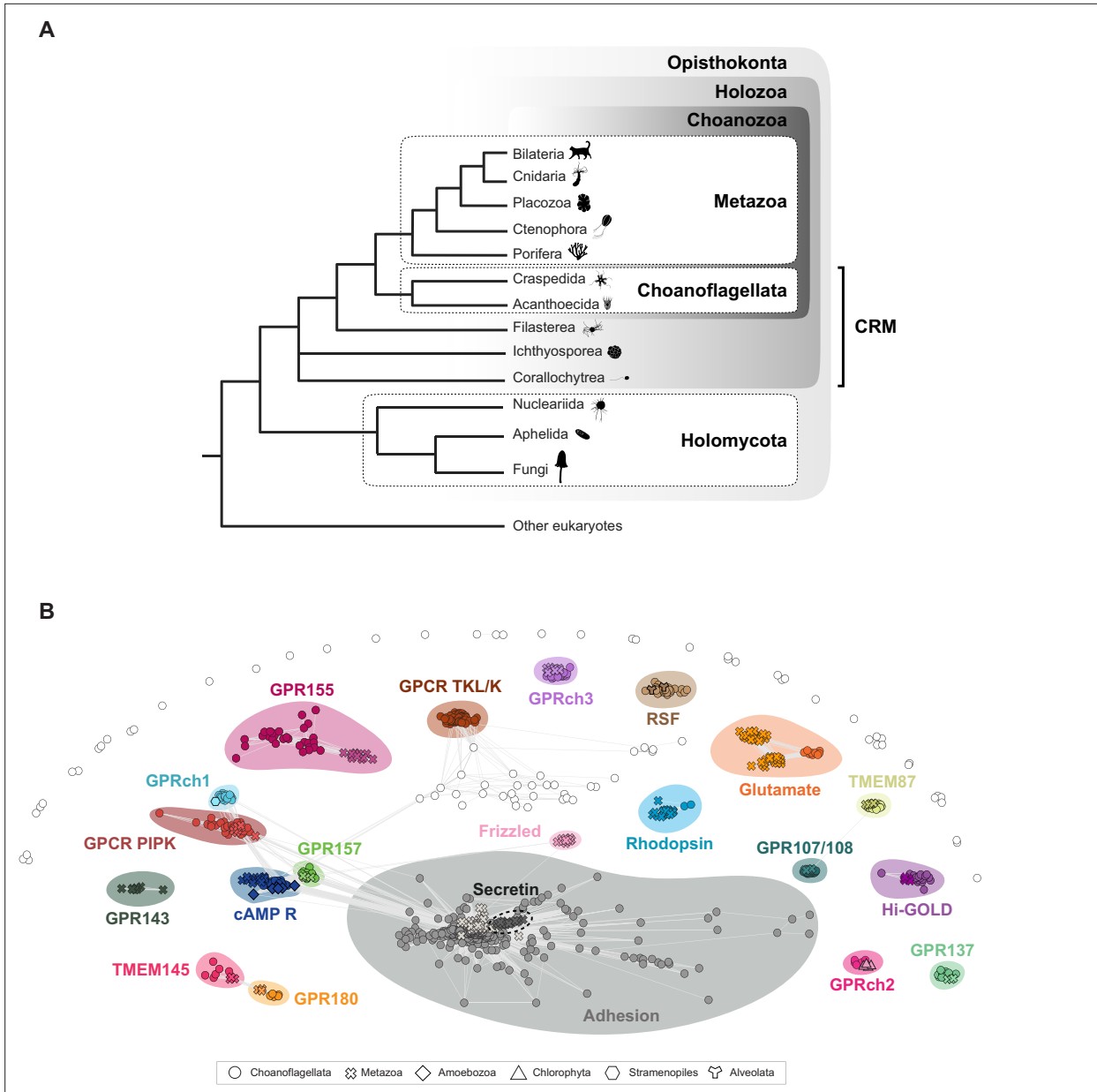

**Figure 1.** Sequence clustering reveals similarities among the 7TMs of G-protein-coupled receptors (GPCRs) from choanoflagellates, metazoans, and other eukaryotes. (**A**) Choanoflagellates are the closest living relatives of metazoans. Shown is a consensus phylogeny of the major lineages analyzed in this study. We use the term 'close relatives of metazoans' (CRM) to denote the paraphyletic group of non-metazoan holozoans that includes choanoflagellates, filastereans, ichthyosporeans, and corallochytreans. Organism silhouettes are from PhyloPic (http://phylopic.org/). (**B**) Most choanoflagellate GPCRs cluster with GPCRs from metazoans and other eukaryotes. The 918 choanoflagellate GPCRs (circles) identified in this study were sorted into clusters based on sequence similarity of their 7TM domains and the 7TM domains of metazoan, amoebozoan, chlorophyte, stramenopile, and alveolate GPCRs. Connecting lines (light gray) correspond to pairwise BLAST scores of p-value <1e⁻⁶. With the exceptions of RSF, GPCR TKL/K, GPRch1, and GPRch2, all choanoflagellate GPCR clusters contained metazoan GPCRs and, in most cases, GPCRs from other eukaryotes. No choanoflagellate GPCRs clustered with metazoan Secretin GPCRs, Frizzled GPCRs, or GPR143 GPCRs. The collection of choanoflagellate GPCRs shown as open circles did not meet the statistical threshold for designation as clusters. All the GPCR sequences used in this analysis are provided in Supplementary file 8.

The online version of this article includes the following figure supplement(s) for figure 1:

**Figure supplement 1.** Pipeline for identifying GPCRs in choanoflagellate genomes and transcriptomes.

**Figure supplement 2.** Total number of GPCRs across choanoflagellates varies by species and seems to correlate with phylogenetic affiliation.

*Figure 1 continued on next page*

*Figure 1 continued*

**Figure supplement 3.** Conservation of metazoan GPCR transducers in choanoflagellates.

**Figure supplement 4.** aGPCRs dominate the GPCRome of most choanoflagellates analyzed.

clusters and 76 additional HMMs for the unclustered GPCRs and used these custom HMMs to re-screen the 23 choanoflagellate proteomes (see Materials and methods: 'Recovering additional choanoflagellate GPCRs using choanoflagellate GPCR BLAST queries and custom choanoflagellate GPCR HMMs'). To complement this approach, we also selected GPCR sequences from each GPCR cluster and used them as BLAST queries to search for additional GPCRs in the 23 choanoflagellate proteomes (see Materials and methods: 'Recovering additional choanoflagellate GPCRs using choanoflagellate GPCR BLAST queries and custom choanoflagellate GPCR HMMs'). After filtering for fragmented sequences, isoforms, and false positives, we recovered 247 additional GPCRs that were not detected during the first round of screening. Together, our bioinformatic analysis identified a total of 918 choanoflagellate GPCRs (Supplementary file 4), with an estimated number of GPCRs per species ranging from 16 in *Barroeca monosierra* to 122 GPCRs in *Acanthoeca spectabilis* (*Figure 1—figure supplement 2*).

To search for downstream signaling components of GPCRs, we aligned the 23 choanoflagellate proteomes with HMMs specific to the three classes of heterotrimeric G proteins (Gα, Gβ, and Gγ; *Neves et al., 2002*; *Dupré et al., 2009*; *Wootten et al., 2018*), positive regulators (Ric8, Phosducin) (*Willardson and Howlett, 2007*; *Srivastava et al., 2019*), and negative regulators (GRK, Arrestin, RGS, GoLoco; *Willard et al., 2004*; *Rajagopal and Shenoy, 2018*; *Bagnato and Rosanò, 2019*; *Gurevich and Gurevich, 2019*; *O'Brien et al., 2019*) of G protein signaling (*Figure 1—figure supplement 1*, *Figure 1—figure supplement 3*, Appendix 1, and Supplementary file 2). Our analysis recovered 141 heterotrimeric G proteins, 308 positive regulators, and 59 negative regulators of G protein signaling (*Figure 1—figure supplement 3*, Appendix 1, and Supplementary files 5-7). Together with the abundant GPCRs detected in choanoflagellates, the finding of nearly complete G protein signaling pathways suggests that choanoflagellates engage in canonical G protein signaling.

To assess the predictive power of our protein-detection pipeline, we then compared the new GPCR and cytosolic signaling component datasets from two choanoflagellates – *Salpingoeca rosetta* and *Monosiga brevicollis* – with previously published GPCR and downstream GPCR signaling component counts for these two species (*Nordstrom et al., 2009*; *Krishnan et al., 2012*; *de Mendoza et al., 2014*; *Krishnan et al., 2015*; *Lokits et al., 2018*). All 11 GPCRs previously described in *M. brevicollis* and all 14 GPCRs found in *S. rosetta* were present in our dataset, along with 10 and 9 newly identified GPCRs, respectively. Similarly, all previously identified heterotrimeric G proteins and diverse positive and negative regulators of G protein signaling from these two choanoflagellates were in our dataset, along with one newly identified Gβ subunit, one RGS, one Goloco, and one additional Arrestin in *M. brevicollis* (*Figure 1—figure supplement 3*; Supplementary file 5). Due to the stringency of our filtering approach and the use of transcriptome-derived proteomes for a majority of choanoflagellate species, these numbers still likely underestimate total GPCR and downstream component abundance and diversity in choanoflagellates. Moreover, in the case of transcriptome-derived proteomes, failure to detect homologs could reflect false negatives rather than a real absence of the gene families. Conversely, sequence contamination and splice isoforms could possibly account for false positives.

## Identification of 18 GPCR families in choanoflagellates

The conservation and alignability of the 7TM domain enable the straightforward categorization of GPCRs into families across eukaryotic diversity (*Attwood and Findlay, 1993*; *Fredriksson et al., 2003*; *Schiöth and Fredriksson, 2005*). Therefore, to investigate the relationships among choanoflagellate GPCRs, we started by extracting and clustering their 7TM sequences. In addition, we included 7TM sequences from diverse metazoans, amoebozoans, stramenopiles, alveolates, and chlorophytes in our analysis to aid in the identification of the choanoflagellate GPCR families (see Materials and methods 'Clustering of the 918 validated choanoflagellate GPCRs'; Supplementary file 8).

Through all-against-all pairwise comparisons of the 7TM domains of the 918 choanoflagellate GPCRs recovered in the previous analysis (*Figure 1—figure supplement 1.3C*) with 7TMs from diverse metazoans, amoebozoans, stramenopiles, alveolates, and chlorophytes, we found that most choanoflagellate GPCRs belong to 18 classes of GPCRs (*Figure 1B*). Six of these classes were previously detected in *S. rosetta* and/or *M. brevicollis*: Glutamate, Adhesion, GPR107/108, cAMP, GPR137,

**Table 1.** List of 18 GPCR clusters identified in choanoflagellates.

| GPCR family | Additional domain(s) | Biological function(s) | Subcellular localization | Natural ligand(s) | References |
|---|---|---|---|---|---|
| Rhodopsin | LRR, LDLa | Light sensing, chemosensing, thermotaxis, energy metabolism, taste (□▲) | Plasma membrane (outer segment, primary cilia), photoreceptor discs, vacuoles (□▲) | Light/retinal, hormones (e.g. GnRH, estrogen, melanocortin), neurotransmitters (e.g. dopamine, acetylcholine), lipids (□▲) | *Cardoso et al., 2012*; *Roy et al., 2020*; *Karthikeyan et al., 2023*; *Chadha et al., 2021*; *Pérez-Cerezales et al., 2015* |
| Adhesion | GPS, EGF, LamG, Calx-beta, HRM, TSP1, EGF_CA, LRR, FN3, Kringle, CUB, MANEC, OLF, SEA, PT, etc. | Cell adhesion, PCP, cell migration, tissue morphogenesis, neurodevelopment, etc. (□) | Plasma membrane (adherens junctions, FAs, dendrites, primary cilia, stereocilia), spindle, nucleus, ER (□) | Proteins of the extracellular matrix (e.g. glycosaminoglycans, collagens, laminin), other membrane proteins (e.g. FLRT, neurexins, teneurins) (□) | *Araç and Leon, 2019*; *Hamann et al., 2015*; *Langenhan et al., 2013*; *Yona et al., 2008*; *Sakurai et al., 2022*; *Kusuluri et al., 2021* |
| Glutamate | ANF, NCD3G, EGF, Cache | Synaptic transmission and neuronal excitability, cell proliferation, cell migration, phototaxis, calcium homeostasis, taste (□) | Plasma membrane (dendrite) (□) | L-Glutamate, GABA, Glycine, sugars, peptide pheromones, etc.(□) | *Laboute et al., 2023*; *Silva and Antunes, 2017*; *Dubovski et al., 2022*; *Qian et al., 2023*; *Sigel and Steinmann, 2012*; *Wong et al., 2022*; *Niswender and Conn, 2010*; *Crupi et al., 2019*; *Elliott and Leys, 2010* |
| cAMP | None | Cell chemotaxis, cell aggregation, cell differentiation, cyst formation (▲) | Plasma membrane (□) | cAMP (▲) | *Klein et al., 1988*; *Saxe et al., 1993*; *Louis et al., 1994*; *Kawabe et al., 2009* |
| GPR137 | None | Cell proliferation, apoptosis, neuronal differentiation (□) | Lysosome (□) | Unknown | *Men et al., 2018*; *Gan et al., 2019* |
| GPR155 | PIN-like transporter, DEP | Nutrient sensing, energy metabolism, tissue patterning (□) | Lysosome (□) | Cholesterol (□) | *Schöneberg, 2024*; *Wang et al., 2018*; *Bayly-Jones et al., 2024*; *Shin et al., 2022* |
| GPR157 | None | Neuronal differentiation (□) | Plasma membrane (primary cilia) (□) | Unknown | *Takeo et al., 2016* |
| GPCR-TKL/K | Tyrosine kinase-like / kinase | Unknown | Unknown | Unknown | This study |
| TMEM145 | GOLD | Structural integrity of hair cell stereocilia (□) | Plasma membrane (stereocilia) (□) | Unknown | *Roh et al., 2025* |
| TMEM87 | GOLD | Protein trafficking, Golgi-Ph maintenance (□) | Golgi (□) | Unknown | *Shin et al., 2020*; *Hirata et al., 2015*; *Hoel et al., 2022* |
| GPR107/108 | GOLD | Protein trafficking, recycling of receptors, immunity (□) | Golgi, nucleus, clathrin vesicles (□) | Neurostatin, gambogic acid (□) | *Balazova et al., 2021*, *Lyu et al., 2022*; *Zhou et al., 2014*; *Yosten et al., 2012*; *Yang et al., 2024* |
| GPR180 | GOLD | Gametogenesis (▲), lipid metabolism (□) | Plasma membrane, vesicles (□) | L-lactate, CTHRC1 (□) | *Mosienko et al., 2018*; *Wang et al., 2022*; *Yoshida et al., 2023* |
| Hi-GOLD | GOLD | Unknown | Unknown | Unknown | This study |
| GPCR PIPK | PIPK | Cell density sensing, bacterial defense, phagocytosis, asexual development, chemotaxis (□) | Endosomal vesicles, phagosome (▲) | Unknown | *Riyahi et al., 2011*; *Hua et al., 2013*; *Bakthavatsalam et al., 2007*; *Bakthavatsalam et al., 2006*; *van den Hoogen et al., 2018* |
| RSF | None | Unknown | Unknown | Unknown | This study |
| GPRch1 | None | Unknown | Unknown | Unknown | This study |

*Table 1 continued on next page*

*Table 1 continued*

| GPCR family | Additional domain(s) | Biological function(s) | Subcellular localization | Natural ligand(s) | References |
|---|---|---|---|---|---|
| GPRch2 | None | Unknown | Unknown | Unknown | This study |
| GPRch3 | None | Unknown | Unknown | Unknown | This study |

□: Metazoa.

▲: Protozoa.

and GPCR_PIPK (*Figures 1B and 2*, *Figure 1—figure supplement 4*; *Table 1*; *Krishnan et al., 2012*; *de Mendoza et al., 2014*; *van den Hoogen et al., 2018*; *Gan et al., 2019*).

This left 12 new GPCR families that had not, to our knowledge, been previously detected in choanoflagellates: Rhodopsin, TMEM145, GPR180, TMEM87, GPR155, GPR157, and six additional GPCR families that appear to fall outside all previously characterized GPCR families in eukaryotes. For reasons that will be discussed further below, we have named these six new GPCR families 'Rémi-Sans-Famille' (RSF), 'Hidden Gold' (Hi-GOLD), GPCR-TKL/K, GPRch1, GPRch2, and GPRch3. (*Figure 1B*; *Table 1*). An additional 76 choanoflagellate GPCRs did not cluster above threshold with any other GPCRs detected from choanoflagellates or other eukaryotes.

## Overview of choanoflagellate GPCR family evolution

Of the 18 GPCR families identified in choanoflagellates (*Table 1*), 16 were found, based strictly on the similarity of their signature 7TM domains, in diverse holozoan and non-holozoan eukaryotes and likely evolved in stem eukaryotes (*Figure 2*; Supplementary file 9). GPR155 was only detected in metazoans and close relatives of metazoans (CRMs; *Figure 1A*), supporting an origin in stem holozoans. The GPCR-TKL/K family was only detected in acanthoecid choanoflagellates, suggesting it evolved in the acanthoecid stem lineage.

Although Adhesion, GPR107/108, TMEM87, TMEM145, Hi-GOLD, RSF, cAMP, GPR137, GPR155, and GPRch3 GPCRs were detected in a wide range of choanoflagellates, other GPCR families appeared to be restricted to one or the other of the two choanoflagellate orders: craspedids (GPR157, Glutamate, GPCR PIPK, and GPRch1) or acanthoecids (GPR180 and GPCR TKL/K; *Figure 2*). The phylogenetic distributions of these GPCR families are likely the result of secondary losses in choanoflagellates because homologs of these GPCRs are detected in other eukaryotes.

Two additional choanoflagellate GPCR families evolved in stem eukaryotes but were lost from metazoans and some CRMs: RSF GPCRs and GPRch1. Frizzled GPCRs, which possibly originated in the last common ancestor of opisthokonts and amoebozoans (*Krishnan et al., 2012*; *de Mendoza et al., 2014*), and GPR143, which emerged in stem holozoans (*Figure 2*; *de Mendoza et al., 2014*), were both lost from stem choanoflagellates.

Metazoan-like Rhodopsins (as defined by GPCRs that are significant matches to the 7tm_1(PF0001) HMM *Clarke and Taylor, 2023*, possess an eight helix *Krishna et al., 2002*; *Kock et al., 2009*; *Knepp et al., 2012*; *Sensoy and Weinstein, 2015*, and present characteristic motifs such as aspartate (D) in TM2, E/DRY in TM3, CWxP in TM6, or NPxxY in TM7 *Krishnan et al., 2012*; *Rinne et al., 2019*) were detected in two choanoflagellate sister species– *Salpingoeca macrocollata* and *Salpingoeca punica* (*Richter et al., 2018*; *López-Escardó et al., 2019*; *Ginés-Rivas and Carr, 2025*). The detection of metazoan-like Rhodopsins in metazoans, choanoflagellates, ichthyosporeans, fungi, and amoebozoans suggests that these receptors predate the split between amoebozoans and opisthokonts (*Figure 2*).

## Metazoan GPCR families that originated in stem eukaryotes

### Glutamate Receptors

The Glutamate Receptor family, which includes important regulators of neuronal excitability, synaptic transmission, and taste recognition in metazoans (*Pin et al., 2003*; *Lagerström and Schiöth, 2008*; *Chun et al., 2012*), first evolved in stem eukaryotes (*Figure 2*; *Krishnan et al., 2012*; *de Mendoza et al., 2014*). In addition to their 7TM domain, Glutamate Receptors have a large, structured extracellular region that varies in different members of the family (*Figure 3A*; *Ellaithy et al., 2020*). For example, mGluR/T1R/CaSR receptors exhibit a bi-lobed ligand-binding domain (ANF) directly

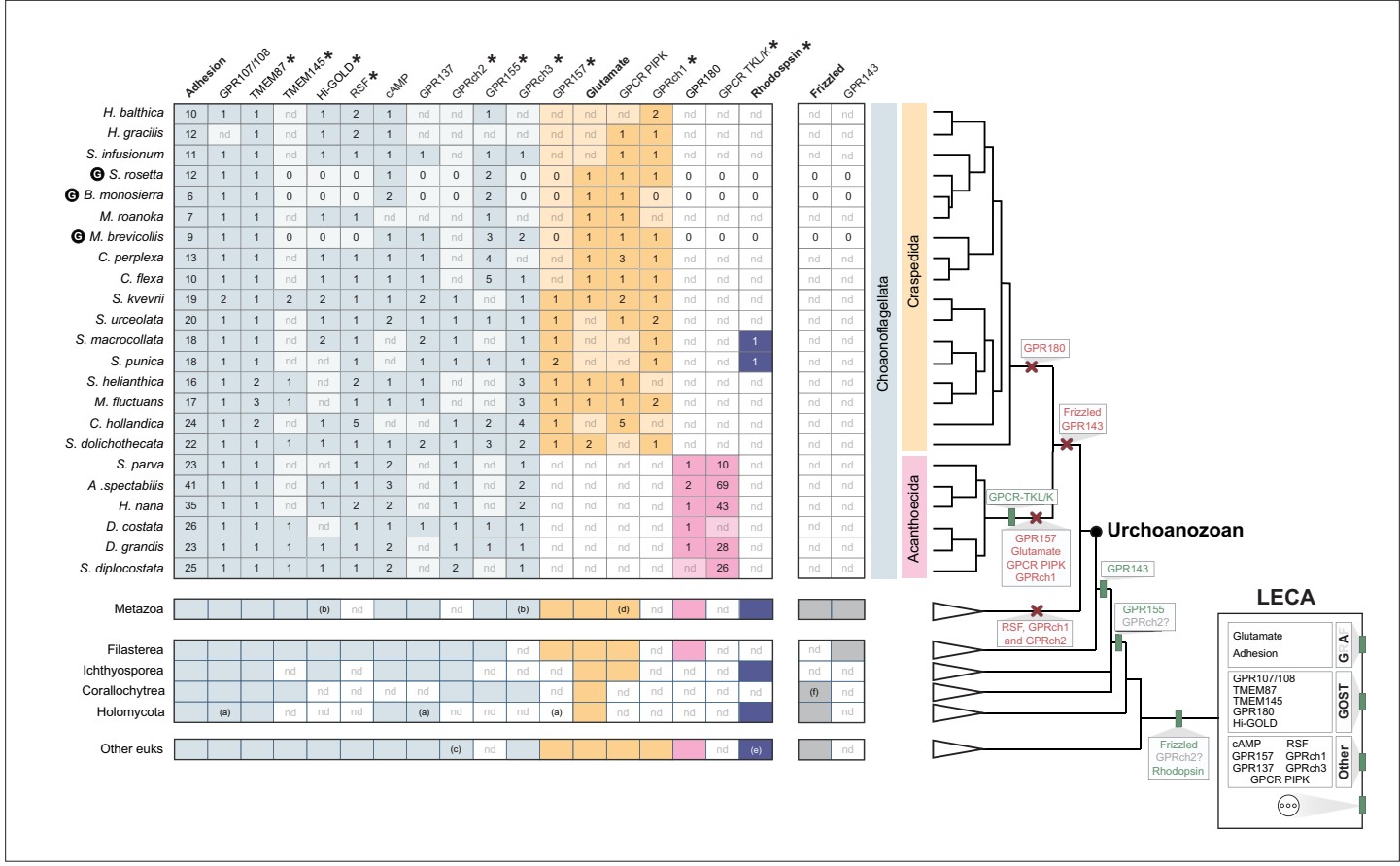

**Figure 2.** Evolutionary history of GPCR families detected in choanoflagellates, metazoans, and other eukaryotes. (**Left**) Taxonomic distribution of GPCR families. Shown in the table are the numbers of GPCR family members (columns) detected in each lineage (rows). The GPCR families are grouped based on their inferred phylogenetic distributions in choanoflagellates: pan-choanoflagellate GPCRs (light blue), GPCR families found in craspedids but not acanthoecids (orange), GPCR families found in acanthoecids but not craspedids (pink), and GPCRs found only in *Salpingoeca macrocollata* and *Salpingoeca punica* (dark blue). Adhesion, Glutamate, Rhodopsin, and Frizzled, four of the five GRAFS GPCR families, are indicated in bold. Asterisks indicate GPCR families that were not known to exist in choanoflagellates prior to this study. For those species in which only transcriptomes are available, gene numbers represent a minimum. nd = not detected in lineages for which only transcriptome data are available. For species with both transcriptome and genome data (*Salpingoeca rosetta*, *Barroeca monosierra*, and *Monosiga brevicollis*; G enclosed within a black circle) (**King et al., 2008**; **Fairclough et al., 2013**; **Hake et al., 2024**), failure to detect a GPCR subfamily member is indicated with a '0'. (**a**) only found in nucleariids, (**b**) lost in vertebrates, (**c**) only found in chlorophytes, (**d**) only found in sponges, (**e**) only found in amoebozoans, (**f**) only found in *Syssomonas multiformis*. (**Right**) A consensus phylogeny shows the relationships among the 23 choanoflagellates included in this study, metazoans, filastereans, ichthyosporeans, corallochytreans, holomycotans, and diverse other eukaryotes. The inferred origins (vertical green rectangle) and subsequent losses (red cross) of GPCR families are indicated at relevant branches on the consensus phylogeny. GPCR families inferred to have originated in the Last Eukaryotic Common Ancestor (LECA) are represented at the root of the phylogeny (box). The presence of additional GPCR families in LECA, not covered in our study, is depicted by three dots. 'GRA' indicates three of the GRAFS GPCR families: Glutamate, Rhodopsin, and Adhesion. 'GOST' indicates subfamilies of GOST GPCRs. Additional GPCR families are listed under 'Other.' Uncertainty about the ancestry of GPRch2 is indicated with a question mark.

The online version of this article includes the following figure supplement(s) for figure 2:

**Figure supplement 1.** GPR157 is an ancient GPCR family conserved in eukaryotes.

**Figure supplement 2.** Predicted structural similarities among metazoan and choanoflagellate GOST family receptors.

**Figure supplement 3.** Features of choanoflagellate GPCR PIPKs.

**Figure supplement 4.** Structural features of GPRch3 and GPRch1 GPCRs.

**Figure supplement 5.** Conservation of metazoan GPR155 in choanoflagellates.

**Figure supplement 6.** Structural features of the newly established RSF GPCR family.

**Figure supplement 7.** Features and evolution of choanoflagellate GPCR TK/TKL/Ks.

**Figure supplement 8.** Structural features of GPRch2 GPCRs.

**Figure supplement 9.** Metazoan Frizzled/Smoothened homologs detected in corallochytreans, fungi, and amoebozoans.

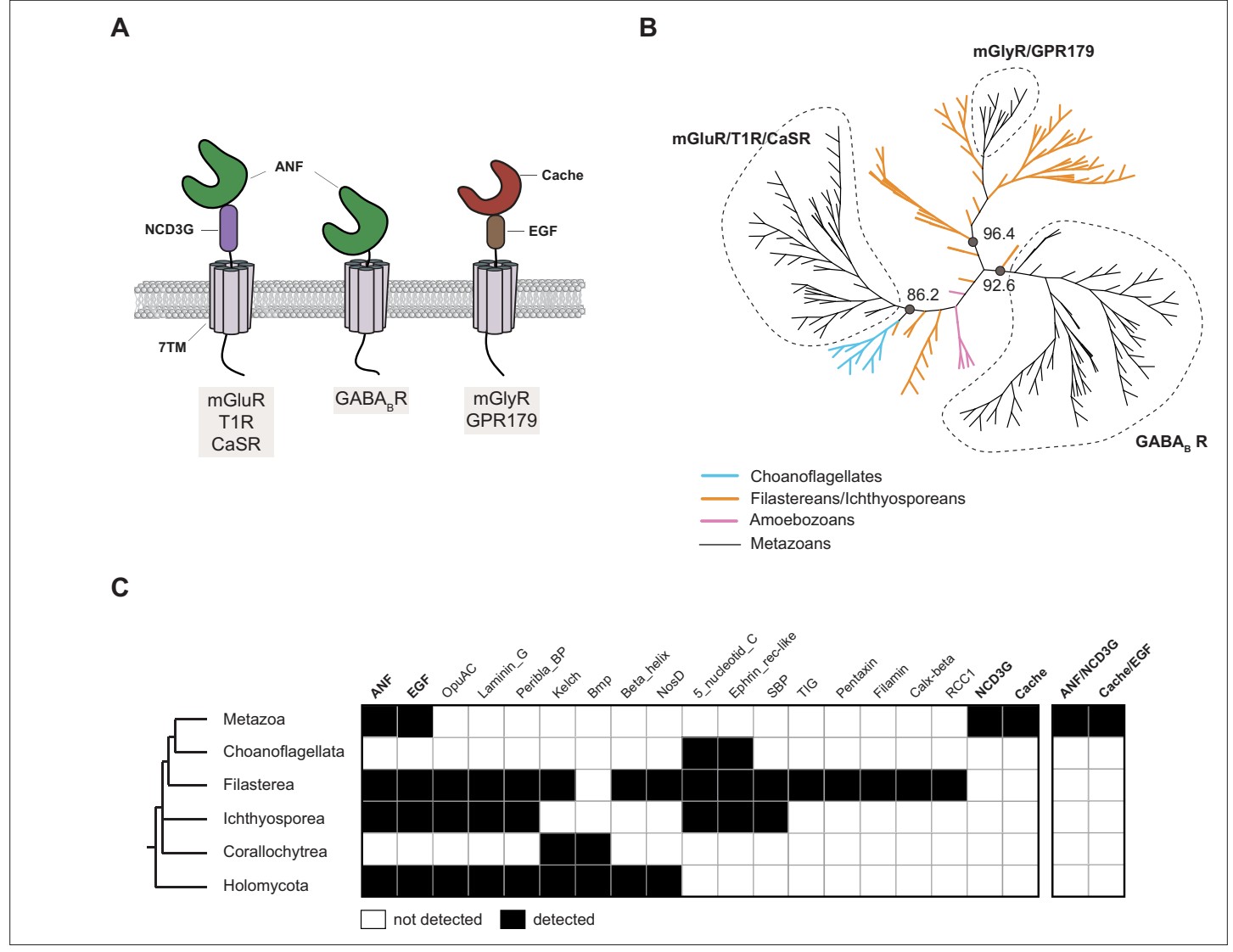

**Figure 3.** Evolution of Glutamate Receptor protein domain architecture. (**A**) Schematic representation of the main protein domains found in metazoan Glutamate Receptor family. GluR/T1R/CaSR receptors have a conserved extracellular module containing an ANF ligand-binding domain fused to a cysteine-rich NCD3G domain. GABA_BR GPCRs contain only the ANF ligand-binding domain. mGlyR/GPR179 GPCRs contain a Cache ligand-binding domain fused to an EGF-like domain. (**B**) Phylogenetic analysis of the 7TM domains of metazoan, choanoflagellate, filasterean, ichthyosporean, and amoebozoan Glutamate Receptors yielded three well-supported clades of GPCRs corresponding to the receptors depicted in panel A. Filasteran and ichthyosporean sequences (orange) clustered as the sister group to metazoan mGluR/CaSR/T1R, mGlyR/GPR179, and GABA_B GPCRs, or branched separately from the rest of GPCRs used in this analysis. Choanoflagellate sequences (light blue) branch as the sister group to the metazoan mGluR/CaSR/T1R GPCRs. No amoebozoan sequence (pink) clustered with metazoan or CRMs. UFboot support values are indicated for each three major nodes of this unrooted maximum-likelihood phylogenetic tree. All the sequences used to build this phylogeny are listed in Supplementary files 10 and 11. (**C**) Phylogenetic distribution of extracellular domains (ECDs) detected in Glutamate Receptors from diverse opisthokonts. While the ANF and EGF protein domains evolved before the diversification of opisthokonts, the presence of the NCD3G and Cache protein domains in GPCRs was not detected in any non-metazoan. Similarly, the ANF/NCD3G and Cache/EGF protein domain modules were only detected in metazoans. Glutamate Receptors from non-metazoans contain diverse extracellular protein domains that are not found in metazoan GPCRs. Protein domains (left) or modules containing pairs of protein domains (right) are indicated in the columns. Taxonomic distribution is indicated in the rows. Protein domains found in metazoan Glutamate Receptors – ANF, EGF, NCD3G, and Cache – are indicated in bold. See Supplementary file 12 for a list of all Glutamate Receptors screened in this analysis.

N-terminal to a cysteine-rich domain (NCD3G) (*Koehl et al., 2019*), both of which are essential for receptor activity (*Hu et al., 2000*; *Jiang et al., 2004*; *Rondard et al., 2006*). In GABA_B receptors, the ANF domain is conserved while the NCD3G is absent (*Evenseth et al., 2020*; *Papasergi-Scott et al., 2020*). Finally, mGlyR/GPR179 receptors present a distinct domain organization, with a ligand-binding

Cache domain sitting atop an EGF-like domain (*Jeong et al., 2021*; *Laboute et al., 2023*; *Rosenkilde and Mathiesen, 2023*; *Yun et al., 2024*).

While members of the Glutamate family and their canonical domain architectures are conserved in a wide range of metazoans, including sponges and ctenophores (*Krishnan et al., 2014*; *de Mendoza et al., 2014*; *Krishnan and Schiöth, 2015*), it is presently unclear how CRM Glutamate Receptors identified by clustering of their 7TM domains relate to their metazoan counterparts. To better understand the evolution of Glutamate Receptors, we conducted a phylogenetic analysis of the 7TM domains from metazoan and CRM Glutamate Receptors (*Figure 3B* and Supplementary files 10 and 11). All choanoflagellate Glutamate Receptors identified in our study formed the sister group to metazoan mGluR/T1R/CaSR GPCRs. The Glutamate Receptors of filastereans and ichthyosporeans branched either at the base of metazoan mGluR/T1R/CaSR receptors, mGlyR/GPR179 receptors, or GABA$_B$ receptors; or at positions on the tree that were distinct from the metazoan Glutamate Receptor families. The Glutamate Receptors from the amoebozan *Dictyostelium discoideum*, of which at least one, GrlE, binds both GABA and Glutamate presumably through its conserved ANF domain (*Anjard and Loomis, 2006*; *Taniura et al., 2006*; *Wu and Janetopoulos, 2013*), grouped separately from metazoan and CRM GPCRs in our analysis.

Next, we assessed the protein domain compositions of the extracellular regions of Glutamate Receptors from diverse opisthokonts (*Figure 3C* and Supplementary file 12). Although earlier studies reported no extracellular domains in choanoflagellate Glutamate Receptors (*Krishnan et al., 2012*; *de Mendoza et al., 2014*), our analysis revealed the presence of 5'-nucleotidase C-terminal domains (5_nucleotid_C) and Ephrin receptor-like domains (Ephrin_rec-like) that are also found in Glutamate Receptors from other CRMs, but not in metazoans.

Two domains present in the N-termini of some metazoan Glutamate Receptors – ANF and EGF – were also found in Glutamate Receptors from other opisthokonts, suggesting an ancient association of these domains with the signature Glutamate 7TM. On the contrary, no non-metazoan GPCRs yet identified possess the canonical metazoan mGluR/T1R/CaSR architecture of an ANF_receptor domain coupled with an NDC3G domain or the mGlyR/GPR179 architecture of a Cache domain coupled with an EGF domain. Together, our data suggest that while the 7TM domains of the three main classes of metazoan Glutamate Receptors – mGluR/T1R/CaSR, mGlyR/GPR179, and GABA$_B$ receptors – have a pre-metazoan origin, the stereotypical combination of domains in their N-terminus was a stem metazoan innovation.

## Rhodopsins

Rhodopsins are the largest GPCR family in most metazoans and include receptors for hormones, neuropeptides, neurotransmitters, nucleotides, and light, among others (*Fredriksson et al., 2003*; *Lagerström and Schiöth, 2008*; *de Mendoza et al., 2014*; *Pándy-Szekeres et al., 2018*). Orthologs of bilaterian Rhodopsins have been found in Cnidaria and Ctenophores but have not been detected in sponges (*Feuda et al., 2014*; *Krishnan et al., 2014*; *Thiel et al., 2023*). While Rhodopsins expanded and diversified in metazoans (*Fredriksson and Schiöth, 2005*; *de Mendoza et al., 2014*; *Thiel et al., 2023*), the origin of this family might have predated the emergence of metazoans, as homologs have been detected in Fungi and in other eukaryotes (*Figure 2*; *Krishnan et al., 2012*; *de Mendoza et al., 2014*).

Upon searching the choanoflagellate genomes and transcriptomes with an HMM profile built solely on the aligned 7TM domains from metazoan Rhodopsin sequences (7TM_1; Supplementary file 2), we identified two previously undetected choanoflagellate Rhodopsins. Additionally, we detected a metazoan-like Rhodopsin from the ichthyosporean *Pirum gemmata* (*Figure 2* and Supplementary file 13).

All-against-all pairwise comparison of the choanoflagellate and ichthyosporean Rhodopsins with the complete Rhodopsin repertoires of representative metazoans (6149 Rhodopsins in total) revealed that choanoflagellate Rhodopsins are most similar to metazoan opsins (including arthropsin, cephalochordate Go opsin, placozoan placopsin, vertebrate opsin 3, and Teleost multiple tissue opsins) and, to a lesser extent, SOG receptors (*Figure 4A and B* and Supplementary files 13 and 14; *Fredriksson et al., 2003*; *Kamesh et al., 2008*; *Nordström et al., 2008*; *McVeigh et al., 2018*; *Yanez-Guerra et al., 2022*). No statistically significant metazoan or choanoflagellate hits were detected for the predicted ichthyosporean Rhodopsin (Supplementary file 14).

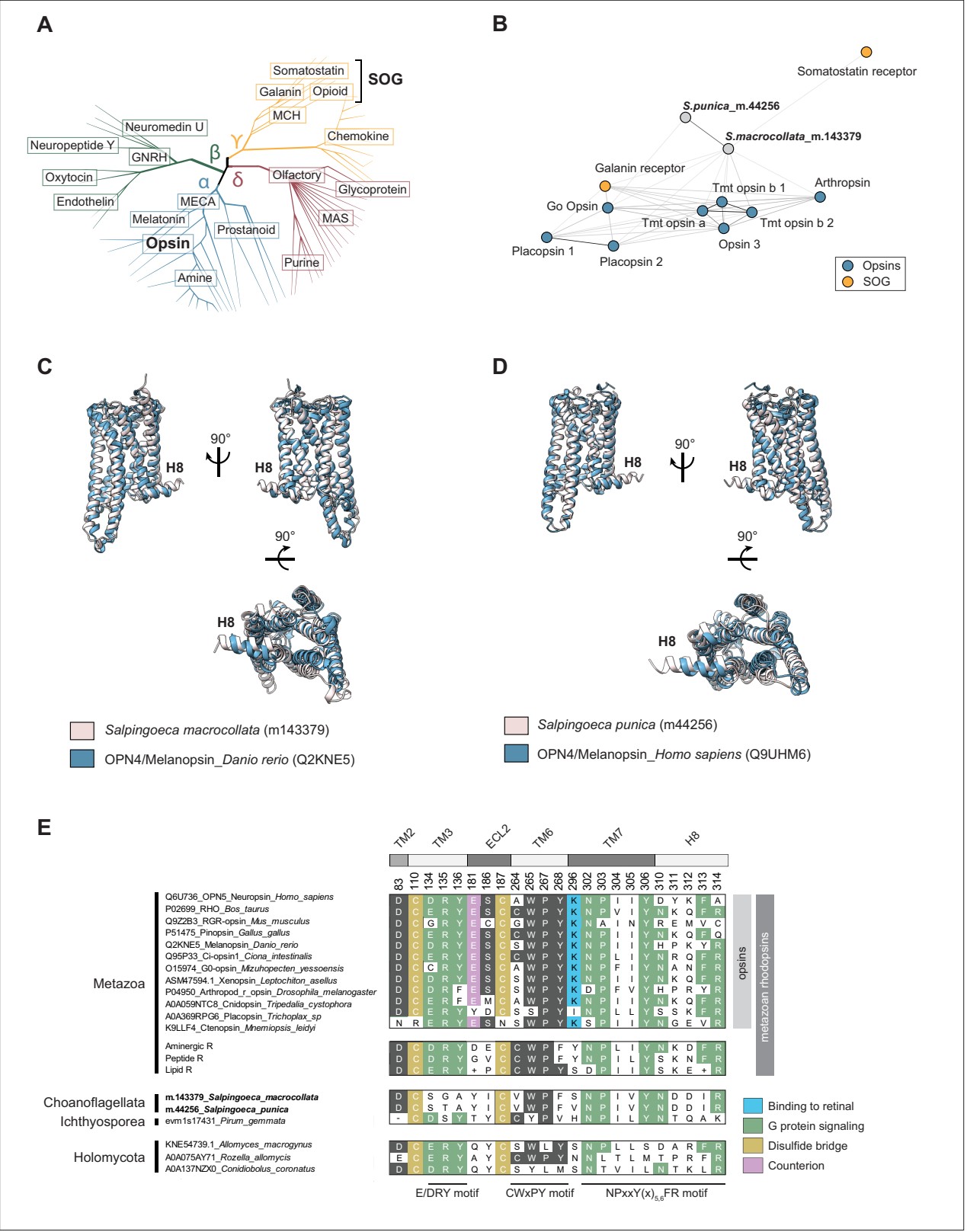

**Figure 4.** Sequence and structural similarities and differences of choanoflagellate Rhodopsins and metazoan opsins. (**A**) Rhodopsin diversity in metazoans; adapted from *Fredriksson et al., 2003*; *Cardoso et al., 2012*; *Lv et al., 2016*. Rhodopsins cluster into four main groups – α (light blue), β (green), γ (yellow), and δ (maroon). The Somatostatin/Opioid/Galanin group of Rhodopsins is indicated as SOG. Opsins and SOG receptors are indicated in bold for reference. (**B**) Choanoflagellate Rhodopsins cluster with opsins and SOG receptors from metazoans. All-against-all pairwise

*Figure 4 continued on next page*

*Figure 4 continued*

comparison of the choanoflagellate Rhodopsins with the complete Rhodopsin repertoire of representative metazoans revealed that choanoflagellate Rhodopsins are most similar to opsins and SOG. Shown is the local sequence similarity network, with nodes indicating Rhodopsin sequences and lines indicating BLAST connections of p-value <1e⁻¹². Choanoflagellate Rhodopsins are shown in gray (*S.macrocollata*_m.143379 and *S.punica*_m.44256), metazoan opsins in blue (Placopsin 1_*T.adherens* (A0A369S8C3), Placopsin 2_*Trichoplax* sp (XP_002114592.1), Tmt opsin a_*D. rerio* (A0A2R8Q4C0), Tmt opsin b 1_*M.mola* (ENSMMOP00000004804.1), Tmt opsin b 2 _*D. rerio* (A0A2R8Q4C0) Opsin_3 *A.platyrhynchos* (XP_012958902.3), Arthropsin 8_*D. pulex* (EFX84032.1), Go Opsin_*B.floridae* (DAC74052.1)), and metazoan SOG receptors in orange (Somatostatin receptor_*B.floridae* (XP_032829507.1), Galanin 1_*H. sapiens* (P47211)). The complete dataset of 6149 Rhodopsins used in this analysis is provided in FASTA format in Supplementary file 13. See Supplementary file 14 for the full analysis. (**C and D**) Protein structure predictions link choanoflagellate Rhodopsins to metazoan opsins. (**C**) The predicted structure of *S. macrocollata* Rhodopsin most closely matches that of opn4a/Melanopsin-A from *Danio rerio*. Shown is the predicted structural similarity of *S. macrocollata* Rhodopsin (m.143379; pink) with Foldseek top hit (E-value: 6.26e⁻¹³) opn4a/Melanopsin-A from *Danio rerio* (AF-Q2KNE5-F1-model_v4; blue). Low confidence regions (>70 pLDDT) were removed for clarity. Shown are views of the superimposed models from the plane of the membrane (top) and from the extracellular perspective (bottom). (**D**) The predicted structure of *S. punica* Rhodopsin most closely matches that of opn4/Melanopsin from humans. Predicted structural similarity of *S. punica* Rhodopsin (m.44256; pink) with Foldseek top hit (E-value: 8.34e⁻¹³) opn4/Melanopsin from humans (AF-Q9UHM6-F1-model_v4; blue). Low confidence regions (>70 pLDDT) were removed for clarity. Shown are views of the superimposed models from the plane of the membrane (top) and from the extracellular perspective (bottom). (**E**) Alignment showing the conservation of functionally important motifs in metazoan, choanoflagellate, ichthyosporean, and holomycotan Rhodopsins. The alignment includes diverse metazoan opsins, three consensus sequences of human non-opsin Rhodopsins (Aminergic, Peptide, and Lipid Rhodopsins; Supplementary file 16), the two choanoflagellate Rhodopsins identified in this study (highlighted in bold), one ichthyosporean Rhodopsin, and three representative Rhodopsins from holomycotans. Residues identified as being critical for Rhodospin protein structure and function are shown. These include: a conserved aspartic acid (**D**) at position 83 in the transmembrane helix 2 (TM2); two conserved cysteines (C; orange) at positions 110 and 187 that are involved in disulfide bond formation; and the conserved E/DRY, CWxPY, and NpxxY(x)$_{5,6}$FR motifs (green), located in TM3, TM6, and TM7/H8, respectively (*Davies et al., 2010*; *Nagata and Inoue, 2021*). These three motifs are essential for G protein interaction and to control the activity of the Rhodopsins. In addition, residues that are specific to opsins are also depicted: Glutamic acid (E)181, which acts as a counterion to the protonated Schiff base (*Davies et al., 2010*; *Hankins et al., 2014*; *Nagata and Inoue, 2021*), Serine (S)186 in extra-cellular loop 2 (EL2), and the highly conserved Lysine (K) at position 296 (blue) in TM7, that is almost universally found across all metazoan opsins (*Gühmann et al., 2022*; *McCulloch et al., 2023*). Lys(K)296 is required for covalent binding to the 11-cis retinal chromophore (*Devine et al., 2013*). Notably, the two choanoflagellate Rhodopsins show a Lys296Ser (*Salpingoeca_macrocollata*_m.143379) and a Lys296Val (*Salpingoeca_punica*_m.44256) substitutions, suggesting that these Rhodopsins may not have light-responsive functions. Canonically conserved functional residues and positions follow bovine Rhodopsin numbering (*Nathans and Hogness, 1983*). The consensus sequences of the three non-opsin subfamilies of human Rhodopsins (Aminergic R, Peptide R, and Lipid R) were downloaded from GPCRdb (https://gpcrdb.org/). (+) symbol is used when consensus sequences cannot be resolved at a given position (ambiguity). The 36 human aminergic receptors, 76 human peptide receptors, and 36 human lipid receptors aligned to build these consensus sequences are provided in Supplementary file 16.

The online version of this article includes the following figure supplement(s) for figure 4:

**Figure supplement 1.** Ichthyosporean Rhodopsin shares structural similarities with metazoan peptide receptor Rhodopsin.

To complement our sequence-based clustering approach, we then predicted the structures of the two choanoflagellate Rhodopsins and the ichthyosporean Rhodopsin using AlphaFold 3 (*Abramson et al., 2024*). We searched for structural matches within three AlphaFold databases: AFDB-PROTEOME, AFDB-SWISSPROT, and AFDB50 (*van Kempen et al., 2024*). In these searches, metazoan opsins (a subclass of Rhodopsins; *Figure 4A*) were recovered as the closest structural matches to choanoflagellate Rhodopsins, with e-values ranging from e⁻¹² to e⁻¹⁴, with Melanopsin/OPN4, an opsin with non-visual and visual functions in metazoans (*Hankins et al., 2008*; *Koyanagi et al., 2013*; *Karthikeyan et al., 2023*), being the top hit for the two choanoflagellate Rhodopsins (*Figure 4C and D*). In contrast, aminergic or peptide Rhodopsin receptors more closely matched the predicted structure of the ichthyosporean Rhodopsin candidate (*Figure 4—figure supplement 1*). Notably, the predicted structures of choanoflagellate and ichthyosporean Rhodopsins exhibit a cytoplasmic helix eight (H8) located immediately after the end of the seventh transmembrane domain, a common feature of metazoan Rhodopsins that is involved in trafficking of the receptor to the cell membrane and in binding to G protein and β-Arrestin (*Figure 4C and D*; *Krishna et al., 2002*; *Kock et al., 2009*; *Knepp et al., 2012*; *Sensoy and Weinstein, 2015*; *Dijkman et al., 2020*).

In metazoan Rhodopsins, conserved sets of amino acids in transmembrane passes 2, 3, 6, and 7 (TM2, TM3, TM6, and TM7), along with extracellular loop 2 (ECL2), serve important structural roles and as determinants of signal transduction (*Figure 4E*, Supplementary files 15 and 16; *Davies et al., 2010*; *Hankins et al., 2014*; *Nagata and Inoue, 2021*). We found that two cysteines (C110 and C187) whose disulfide bridges often connect ECL2 and TM3 in metazoan Rhodopsins (*Rader et al., 2004*) were conserved in both the choanoflagellate and ichthyosporean receptors. Similarly, asparagine D83 was conserved in TM2 of choanoflagellate and holomycotan Rhodopsins. Among

the sites shown to be important for controlling G protein signaling, the CWxPY and NpxxY(x)$_{5,6}$FR motifs were partially conserved in choanoflagellate and ichthyosporean sequences. In contrast, the E/DRY motif, which might regulate the activation of metazoan Rhodopsins (*Rovati et al., 2007*; *Sandoval et al., 2016*), was lost from TM3 in the choanoflagellate Rhodopsins but partially conserved in the ichthyosporean Rhodopsin. Finally, none of the residues mediating light sensing in opsins – namely the highly conserved Lysine K269, which mediates the binding to the chromophore retinal and the glutamic acid E181 that serves as a counterion to the protonated Schiff base (*Davies et al., 2010*; *Hankins et al., 2014*; *Nagata and Inoue, 2021*) – were detected in CRM or holomycotan Rhodopsins, suggesting that these residues (and their functions in light sensing) evolved in stem metazoans.

## Adhesion GPCRs

In metazoans, Adhesion GPCRs (aGPCRs) are generally the second most abundant GPCR family after the Rhodopsin family (*Kamesh et al., 2008*; *Nordström et al., 2008*; *Krishnan et al., 2014*; *de Mendoza et al., 2014*). They are critical for the multicellular biology of metazoans, in which they regulate epithelial morphogenesis, neuronal development, and immunity and are implicated in the progression of various cancers (*Simundza and Cowin, 2013Hamann et al., 2015*; *Langenhan et al., 2016*; *Liebscher et al., 2022*). The signature 7TM domain of aGPCRs has been detected in CRMs, holomycotans, and diverse other lineages (*Figure 2*; *Krishnan et al., 2012*; *de Mendoza et al., 2014*).

Phylogenetic analysis of the 7TM domains of choanoflagellates uncovered at least 19 subfamilies of aGPCRs (subfamilies α-$\tau$ ; *Figure 5—figure supplement 1A and B*; Supplementary files 17 and 18), of which only one, class θ, is orthologous to a metazoan aGPCR subfamily (Bootstrap support 83%; *Figure 5A* and Supplementary files 19 and 20), the ADGRV family (*Weston et al., 2004*; *Hamann et al., 2015*; *Scholz et al., 2019*; *Kusuluri et al., 2021*). The other choanoflagellate aGPCR families presumably evolved on the choanoflagellate stem lineage or within choanoflagellates, as they are not detected in other lineages. In addition, a set of uncharacterized cnidarian and cephalochordate GPCRs grouped with a subset of filasterean aGPCRs (Bootstrap support 98%, *Figure 5A*).

Unlike other GPCRs, metazoan aGPCRs have a large N-terminal extracellular region with diverse protein domains that are linked to the 7TM domain by a conserved GPCR-autoproteolysis-inducing (GAIN) domain (*Figure 5B*; *Araç et al., 2012*; *Langenhan et al., 2013*; *Hamann et al., 2015*). Ligand binding can cause the N-terminal fragment (NTF) to be released from the rest of the protein at the GPCR proteolytic site (GPS), exposing a tethered agonist element (TA) that activates downstream G protein signaling (*Barros-Álvarez et al., 2022*; *Kleinau et al., 2023*; *Ping et al., 2022*; *Xiao et al., 2022*; *Seufert et al., 2023*). The structural complexity of the metazoan NTF, including the GAIN domain and additional extracellular domains (ECDs), appears restricted to holozoans, as holomycotans and other eukaryotes have shorter, unstructured N-termini (*Krishnan et al., 2012*). Therefore, a detailed analysis of aGPCR subfamilies and their protein domains in CRMs promised to illuminate the evolution of aGPCRs after the holozoan-holomycotan divergence.

Although the phylogeny of aGPCRs in choanoflagellates was inferred solely based on the comparison of their 7TM domains, the resulting groups tended to cluster receptors that also shared sequence similarity and protein domain architecture in their NTFs (*Figure 5C*, *Figure 5—figure supplement 2A*). The connection between the 7TM phylogeny and the N-terminal protein domain architectures of different aGPCRs, which was previously reported for metazoan aGPCR families (*Bjarnadóttir et al., 2007*; *Hamann et al., 2015*), is exemplified by the choanoflagellate aGPCR groups α, $\varepsilon$, $\zeta$ , θ, $\nu$ , and $o$ , whose NTFs generally contain one or multiple conserved protein domains that are shared among members (*Figure 5C*). Comparison of metazoan and CRM NTFs revealed that only ADGRV receptors share robust N-terminal sequence similarity among metazoan, choanoflagellate, and filasterean aGPCRs (*Figure 5—figure supplement 3* and Supplementary file 21).

While the GAIN and aGPCR 7TM domains evolved before the origin of opisthokonts (*Araç et al., 2012*; *Krishnan et al., 2012*; *de Mendoza et al., 2014*), we detected the fusion of these two domains into a single module (GAIN/7TM) in most, but not all, holozoan aGPCRs (*Figure 5D*, *Figure 5—figure supplement 2B* and *Figure 5—figure supplement 4A*; Supplementary file 22; *Prömel et al., 2013*; *Krishnan et al., 2014*). Therefore, the GAIN/7TM module likely evolved in stem holozoans. Indeed, motifs necessary for self-proteolysis in metazoan GAIN domains (*Araç et al., 2012*; *Stoveken et al., 2015*; *Kleinau et al., 2023*; *Ping et al., 2022*; *Xiao et al., 2022*; *Seufert et al., 2023*) are also

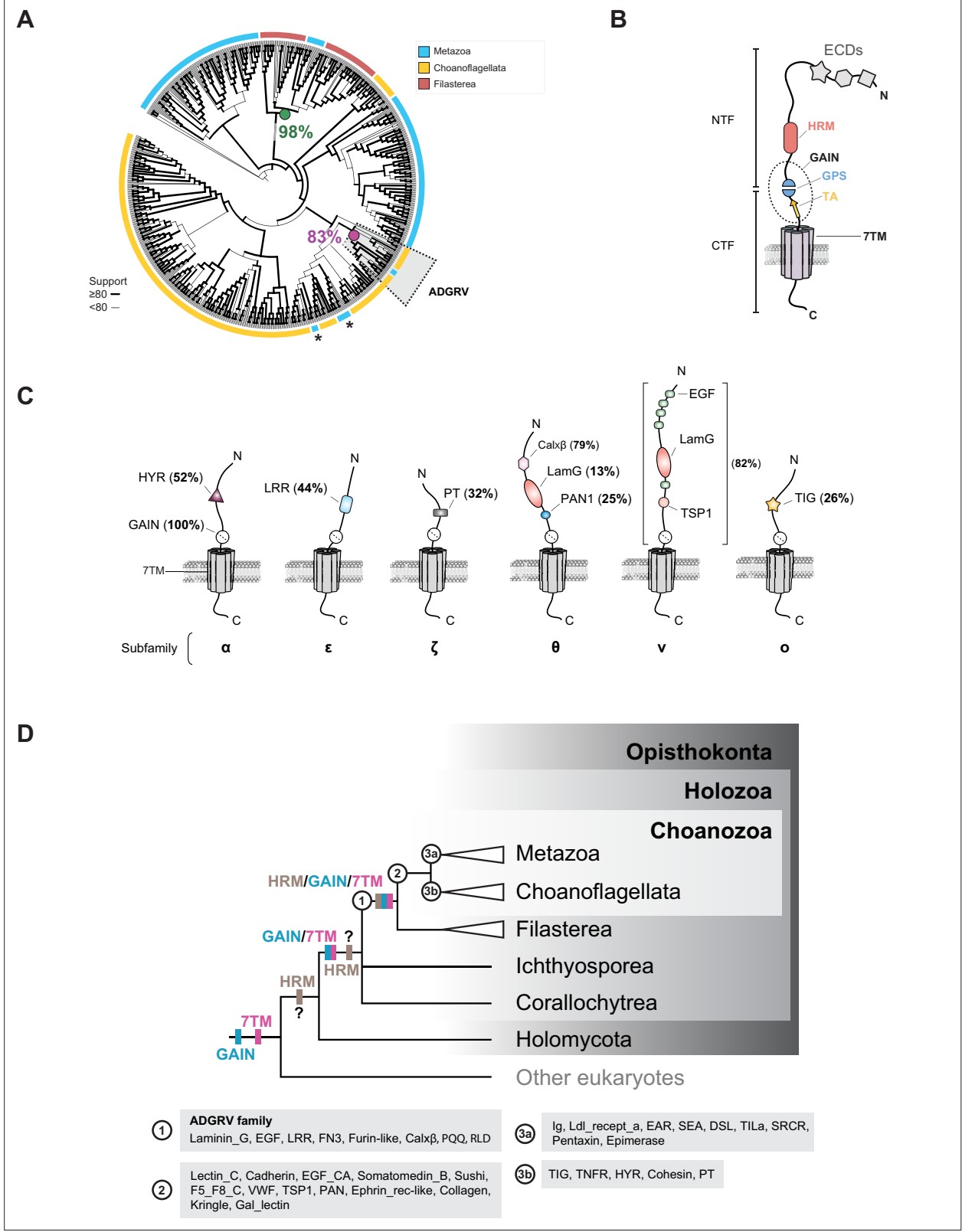

**Figure 5.** Evolution of aGPCR protein domain architecture. (**A**) Although aGPCRs were present in LECA (*Figure 2*), phylogenetic analysis of holozoan aGPCRs revealed that they diversified independently in metazoans and CRMs. Most metazoan (blue), choanoflagellate (orange), and filasterean (red) aGPCRs formed distinct clades in our analysis, suggesting an absence of orthologous relationships between the 7TM region of the aGPCRs from these three clades. A notable exception was the metazoan **AD**hesion **G**-protein-coupled **R**eceptor **V** (ADGRV) GPCRs that grouped with choanoflagellate 7TM

*Figure 5 continued on next page*

*Figure 5 continued*

sequences (dotted gray box; 83% bootstrap support for ancestral node, magenta circle), suggesting they are orthologous. In addition, we also observed that members of the metazoan ADhesion G-protein-coupled Receptor A (ADGRA) subfamily (asterisks) tended to group with choanoflagellate aGPCRs but either lacked reliable confidence value support or were not systematically recovered in all the inferred phylogenies. A subset of filasterean aGPCRs clustered with a set of uncharacterized cnidarian and cephalochordate receptors (bootstrap support 98%, green circle). This maximum-likelihood phylogenetic tree infers the evolutionary history of the 7TM domain of 329 choanoflagellate aGPCRs identified in this study, along with 76 filasterean aGPCRs and 253 metazoan aGPCRs. The 7TM sequence of a ciliate aGPCR (Stentor_coeruleus_OMJ80129.1_19670) was also included in the analysis and used as an outgroup to root the tree. The width of branches scales with UFboot support for the ancestral node. Branch lengths do not scale with evolutionary distance in this rendering. All the 7TM sequences used in this phylogenetic reconstruction are found in Supplementary file 19, and the fully annotated version of this phylogenetic tree, including bootstrap values, branch lengths, and all species names, is found in Supplementary file 20. (**B**) Protein domain architecture of metazoan aGPCRs. Like all GPCRs, aGPCRs contain a 7TM domain (represented by seven barrels) that anchors the protein in a lipid bilayer. What separates aGPCRs from other GPCRs is their possession of an autoproteolytic GAIN domain (dotted oval) containing a proteolysis site (GPS, blue) and a hydrophobic tethered agonist element (TA, orange). The cleavage site in the GPS represents the boundary between the N-terminal fragment (NTF) and the C-terminal fragment (CTF). Many, but not all, aGPCRs also contain an HRM domain (salmon) within a few hundred amino acids of the GPS (*Figure 5—figure supplement 5C*; *Prömel et al., 2012*; *Araç et al., 2016*). In metazoans, the NTF often contains a diversity of additional extracellular protein domains (ECDs, represented by gray star, hexagon, and square) that likely contribute to the diversity of ligands bound by aGPCRs (*Araç and Leon, 2019*; *Knierim et al., 2019*). (**C**) Diversity and conservation of protein domains in the NTFs of different choanoflagellate aGPCR families. Like metazoan aGPCRs, nearly all choanoflagellate aGPCRs contain a GAIN domain with conserved GPS and TA (*Figure 5—figure supplement 4*). Many aGPCR subfamilies identified through their 7TM were also distinguishable through their characteristic combinations of N-terminal protein domains (*Figure 5—figure supplement 2A*). Shown here are six subfamilies – $\alpha$, $\varepsilon$, $\zeta$, $\theta$, $\nu$, and $o$ – whose members frequently share one or more protein domains. While subfamilies $\alpha$, $\varepsilon$, $\theta$, $\nu$, and $o$ are seemingly unique to choanoflagellates, subfamily $\theta$ is notable because the full-length proteins (both the NTFs and CTFs) are homologous to those of the ADGRV subfamily in metazoans (*Figure 5A*, *Figure 5—figure supplement 3*). Percentages indicate the number of members within a given subfamily possessing a conserved protein domain (subfamilies $\alpha$, $\varepsilon$, $\zeta$, $\theta$, and $\xi$) or a conserved combination of protein domains (subfamily $\nu$). All the choanoflagellate aGPCR NTF sequences used in these analyses are found in Supplementary file 21 (**D**) Hypothesized evolution of the HRM/GAIN/7TM module and additional ECDs in aGPCRs. For four nodes on the eukaryotic tree of life – (1) the last common ancestor of filastereans and choanozoans, (2) stem choanozoans, (3 a) stem metazoans, and (3b) stem choanoflagellates – we reconstructed the phylogenetic distribution of diverse protein domains in the NTFs of aGPCRs. Most aGPCRs in metazoans, choanoflagellates, and filastereans contain a protein domain module composed of a GAIN domain and a 7TM. Additionally, the HRM/GAIN/7TM module, which is less common than the GAIN/7TM module in aGPCRs, is nonetheless conserved across most metazoan, choanoflagellate, and filasterean lineages. The linkage of the GAIN and 7TM domains likely occurred in stem holozoans, as today the two domains are restricted to the aGPCRs of extant holozoans, although the two domains are encoded in other genes in diverse non-holozoans (*Araç et al., 2012*; *Krishnan et al., 2012*; *de Mendoza et al., 2014*), suggesting that they originated in earlier branches of the eukaryotic tree. The linkage of the HRM domain with the GAIN/7TM module likely occurred in the last common ancestor of filastereans and choanozoans. The HRM domain is largely restricted to holozoans, although we found six non-aGPCR proteins containing HRM domains in the proteome of the nucleariid *Fonticula alba* (*Figure 5—figure supplement 5D*). Therefore, HRMs either evolved in stem holozoans and were incorporated into the nucleariids by horizontal gene transfer, they are homologous in the two lineages and were lost from most holomycotans, or the similarities of HRMs in *F. alba* and holozoans are the result of convergent evolution. The repertoires of ECDs inferred for each node – (1), (2), (3 a), and (3b) – are shown in gray boxes labeled accordingly. Finally, ADGVR likely evolved in the last common ancestor of filastereans and choanozoans.

The online version of this article includes the following figure supplement(s) for figure 5:

**Figure supplement 1.** Diversity of aGPCR subfamilies in choanoflagellates.

**Figure supplement 2.** Diversity and evolution of aGPCR extracellular protein domains in choanoflagellates and other opisthokonts.

**Figure supplement 3.** Independent diversification of most aGPCR NTFs in metazoans and CRMs.

**Figure supplement 4.** Conservation of the GAIN domain in metazoans, choanoflagellates, filastereans, and corallochytreans.

**Figure supplement 5.** Pre-metazoan origin of the HRM domain and HRM-GAIN-7TM module.

conserved in CRM aGPCRs, suggesting that these receptors may similarly undergo cleavage and activation (*Figure 5—figure supplement 4C*; Supplementary file 23).

The presence of other aGPCR ECDs (*Figure 5B*) likely evolved in the last common ancestor of filastereans and choanozoans. This is supported by the absence of ECDs from ichthyosporean, corallochytrean, or non-holozoan aGPCRs (*Figure 5—figure supplement 2B*), coupled with the conservation of Laminin_G, EGF, LRR, FN3, Furin-like, Calxβ, PQQ, and RLD domains in the aGPCRs of filastereans with those of either choanoflagellates or metazoans (*Figure 5D*, node '1'). We infer that the aGPCR ECD repertoire diversified in the choanozoan stem lineage (*Figure 5D*, node '2'), followed by lineage-specific acquisitions of additional ECDs in metazoans and choanoflagellates (*Figure 5D*, node '3 a' and '3b'; *Figure 5—figure supplement 2B*). Interestingly, the HRM domain — previously identified only in the metazoan Secretin and aGPCR families (*Figure 5—figure supplement 5A* and *Lagerström and Schiöth, 2008*; *Nordstrom et al., 2009*; *Krishnan et al., 2012*; *Krishnan et al., 2014*; *Araç et al.,*

*2016*) — was also found in 22 choanoflagellate and 6 filasterean aGPCRs in our analysis (*Figure 5—figure supplement 5B*). Like its positioning in metazoan aGPCRs (*Araç et al., 2012*; *Prömel et al., 2012*), the HRM domain is located directly N-terminal to the GAIN domain in choanoflagellate and filasterean aGPCRs, with a comparable separation of ~280–330 amino acids between the C-terminus of the HRM domain and the N-terminus of the 7TM domain (*Figure 5—figure supplement 5C*). These findings suggest that the HRM domain and its integration into an HRM-GAIN-7TM module likely originated in the last common ancestor of Filasterea and Choanozoa (*Figure 5D*). Supporting this view, the HRM domain was absent from all non-holozoan proteomes analyzed, with one exception: the nucleariid *Fonticula alba* (*Figure 5—figure supplement 5D*), a close relative of Fungi (*Liu et al., 2009*; *Torruella et al., 2015*). *Fonticula alba* encodes five non-GPCR transmembrane proteins that contain HRM domains (Supplementary file 24).

These findings clarify the evolution of key features of holozoan aGPCRs. While the autoproteolytic GAIN domain (*Araç et al., 2012*), the aGPCR 7TM domain (*Krishnan et al., 2012*; *de Mendoza et al., 2014*), and possibly the HRM domain (this study) existed before the emergence of Holozoa, our analysis indicates that they were first combined into a single module in holozoans. The GAIN-7TM module originated in stem holozoans, followed by the positioning of the HRM domain N-terminal to the GAIN domain in the last common ancestor of Filasterea and Choanozoa. This was followed by the acquisition of additional extracellular protein domains in choanozoans and filastereans (*Figure 5D*).

## cAMP Receptors

Twenty-eight choanoflagellate GPCRs grouped with cAMP Receptors from diverse other eukaryotes (*Figure 1B*). These receptors possess short N- and C-termini and a unique 7TM signature that distinguishes them from other GPCRs (*Saxe et al., 1993*; *Krishnan et al., 2012*; *Greenhalgh et al., 2020*).

The cAMP Receptors were present in stem eukaryotes and are inferred to have given rise to GRAFS GPCRs (*Nordstrom et al., 2011*; *Krishnan et al., 2012*). cAMP Receptors were first described in the amoebozoan *Dictyostelium discoideum*, where they control chemotaxis, aggregation, morphogenetic movements, cell growth, and many developmentally regulated genes (*Klein et al., 1988*; *Johnson et al., 1993*; *Saxe et al., 1993*; *Louis et al., 1994*). While cAMP Receptors bind to cAMP in *D. discoideum* (*Juliani and Klein, 1981*; *Caterina et al., 1995*; *Milne et al., 1997*), the precise cAMP binding site(s) have yet to be experimentally identified (*Greenhalgh et al., 2020*). Since then, cAMP Receptors have been identified in holozoans, holomycotans, alveolates, plants, and other lineages (*Figure 2*; *Krishnan et al., 2012*; *de Mendoza et al., 2014*). Notably, although present in diverse metazoans, cAMP Receptors have been lost from vertebrates. Previous studies found that the choanoflagellates *S. rosetta* and *M. brevicollis* each encode a single cAMP Receptor (*Krishnan et al., 2012*; *de Mendoza et al., 2014*). This study found cAMP Receptors in 17 additional choanoflagellate species, including craspedids and acanthoecids (*Figure 2*).

## GPR137

GPR137 is a lysosomal receptor that controls mTORC1 translocation to the lysosomes and modulates the Hippo pathway (*Gan et al., 2019*). Moreover, GPR137 regulates autophagy, cell growth, and cell proliferation and is implicated in various cancers (*Men et al., 2018*; *Iwasa et al., 2023*; *Li et al., 2024*). GPR137 does not possess other protein domains outside the 7TM domain. The receptor is broadly conserved in metazoans, and homologs have been found in the choanoflagellate *Monosiga brevicollis* and the amoebozoan *D. discoideum* (*Figure 2* and *Gan et al., 2019*).

We detected GPR137 in a wide range of choanoflagellate species, as well as in filastereans, ichthyosporeans, and non-holozoans, including nucleariids and other eukaryotic lineages (*Figure 2*). Thus, GPR137 probably evolved at the stem of the eukaryotic tree. Whether GPR137 controls cell growth and the cell cycle in non-metazoans remains to be tested.

## GPR157

GPR157 Receptors promote neuronal differentiation of radial glial progenitors in mice, where they localize to the primary cilia and signal via a G(q)-protein that activates a phosphatidylinositol-calcium second messenger (*Takeo et al., 2016*). No additional domains outside the 7TM have been detected in GPR157.

GPR157 had previously been identified only in mammals (*Takeo et al., 2016*). In this study, we detected GPR157 homologs in most metazoans, including sponges, as well as in choanoflagellates, filastereans, nucleariids, amoebozoans, rhizarians, and various other eukaryotes, suggesting that GPR157 likely evolved at the root of the eukaryotic tree (*Figure 2*, *Figure 2—figure supplement 1A*; Supplementary file 25). Choanoflagellate and metazoan GPR157 share a level of sequence similarity with the 7TM domains of cAMP, aGPCRs, and GPCR PIPKs (*Figure 1B*). In addition, we noticed a cytoplasmic helix 8 (H8), better known from Rhodopsins, directly C-terminal to TM7 in choanoflagellate, metazoan, and other eukaryote GPR157 protein structure predictions (*Figure 2—figure supplement 1B*). Striking conservation of motifs that have not previously been characterized was noted in TM2 ('LLxxLSL/V/IxD'), ECL2 ('WCWI/V/L'), and TM7 ('QGxxNxIxF') of the GPR157 TM domain (*Figure 2—figure supplement 1C*).

## GPR107/108, TMEM145, GPR180, and TMEM87 (GOST GPCRs)

We identified 63 choanoflagellate GPCRs that clustered with a group of metazoan GPCRs named 'GOST' for Golgi-dynamics domain seven-transmembrane helix proteins (*Figure 1* and *Table 1*; *Hoel et al., 2022*). All GOST members exhibit a Golgi dynamics (GOLD) domain in their extracellular region, which is possibly involved in ligand recognition, followed directly by the 7TM domain (*Anantharaman and Aravind, 2002*; *Hoel et al., 2022*). GOST GPCRs include GPR107/108, TMEM145, GPR180, and TMEM87 – all found in choanoflagellates – along with GPR181 and Wntless, neither of which was detected in choanoflagellates.

The structural similarity between choanoflagellate and metazoan GOST receptors includes both the 7TM domain and the N-terminal GOLD domain (*Figure 2—figure supplement 2*). Nonetheless, it is unclear whether the GOST GPCRs are monophyletic, as GPR107/108/TMEM87 and GPR180/TMEM145 grouped separately in our analyses of the 7TM domains (*Figure 1B*). Despite this uncertainty, all GOST members have homologs in holozoans and various other eukaryotes (*Figure 2*), suggesting that they were present in stem eukaryotes.

GOST GPCRs are understudied, but the characterization of several members showed that they localize at the Golgi and endocytic vesicles, suggesting a common function in trafficking membrane-associated cargo (*Zhou et al., 2014*; *Hirata et al., 2015*; *Shin et al., 2020*; *Hoel et al., 2022*; *Mitrovic et al., 2024*). In addition, a recent study found that TMEM145 also contributes to the structural integrity of hair cell stereocilia (*Roh et al., 2025*), expanding the functions associated with GOST GPCRs. Although ligands have been proposed for some GOST proteins– neurostatin for GPR107 (*Yosten et al., 2012*; *Yang et al., 2024*), gambogic acid for GPR108 (*Lyu et al., 2022*), L-lactate (*Mosienko et al., 2018*), and CTHRC1 for GPR180 (*Balazova et al., 2021*)– it is presently unclear whether these receptors are competent for G protein signaling since no molecular evidence has been reported for any member of this family (*Balazova et al., 2021*; *Hoel et al., 2022*).

## Hi-GOLD

Seventeen choanoflagellate GPCRs clustered independently from other known GOST GPCRs but exhibit a similar topology with a conserved GOLD domain directly N-terminal to their 7TM (*Figure 1B*, *Figure 2—figure supplement 2*). We named this group of GPCRs 'Hidden GOLD' (Hi-GOLD). Hi-GOLD GPCRs are conserved in both craspedids and acanthoecids (*Figure 2*). Additionally, they have homologs in metazoans, including sponges, ctenophores, placozoans, cnidarians, holothuroideans, and cephalochordates, but are absent from vertebrates (Supplementary file 26). Hi-GOLD GPCRs were also detected in other CRMs (filastereans and ichthyosporeans) and non-holozoans, such as amoebozoans. Thus, we have uncovered a new group of GPCRs bearing structural similarity with GOST GPCRs and having ancient origins in eukaryotes.

## GPCR PIPK

Twenty-seven choanoflagellate GPCRs were identified as GPCR PIPKs based on their domain composition. These receptors comprise an N-terminal 7TM combined with a C-terminal phosphatidylinositol phosphate kinase (PIPK) catalytic domain (*Bakthavatsalam et al., 2006*; *van den Hoogen et al., 2018*; *van den Hoogen and Govers, 2018*; *Figure 2—figure supplement 3A and B*). A diagnostic 'LR(x)₉GI' motif, previously found to be present in the linker region between the 7TM domain and the

PIPK domain of these receptors (*van den Hoogen et al., 2018*) was also found in the choanoflagellate GPCR PIPKs (*Figure 2—figure supplement 3C*).

Initially discovered in oomycetes, where the family greatly expanded, GPCR PIPKs have now been detected in diverse eukaryotes and sponges (*Bakthavatsalam et al., 2006*; *van den Hoogen et al., 2018*). Sponges appear to be the only metazoans encoding GPCR PIPKs (*Figure 2* and *van den Hoogen et al., 2018*). Thus, genes encoding GPCR PIPKs were likely present in the last common eukaryotic ancestor and secondarily lost in diverse eukaryotes, including a loss after the divergence of sponges from the rest of metazoans.

A previous study found that one GPCR PIPK each was encoded in *S. rosetta* and *M. brevicollis* species *van den Hoogen et al., 2018*; our analysis revealed that GPCR PIPKs are also expressed by most craspedid choanoflagellates (*Figure 2*). However, no acanthoecid choanoflagellates in our study appeared to possess these receptors (*Figure 2*).

While GPCR PIPKs have previously been grouped into the same family because of their domain architecture– the 7TM domain combined with a PIPK domain– our analysis suggests that this family is also supported by the sequence homology of their 7TM domain. Indeed, both choanoflagellate and sponge GPCR PIPKs analyzed in this study clustered together, forming a class distinct from other GPCR families (*Figure 1B*). In addition, blasting the 7TM domains of choanoflagellate GPCR PIPKs against the full EUKPROT database (*Richter et al., 2022*) recovered GPCR PIPKs from various other eukaryotes (Filasterea, Discoba, Collodictyonida, Telonemida, Rhodelphida, Ancyromonadida, and Centroplasthelida), suggesting that the 7TM domain is enough to find homologs, independently of the C-terminal PIPK domain. Thus, GPCR PIPKs likely constitute a family of GPCRs on their own, with a signature 7TM domain that might have evolved in stem eukaryotes.

Experimental data regarding the function of GPCR PIPKs is limited. Pioneer studies in oomycetes implicated the receptor in the regulation of sexual reproduction and virulence (*Hua et al., 2013*). In contrast, studies of *D. discoideum* GPCR PIPKs suggested a role in phagocytosis, cell density sensing, and bacterial defense (*Bakthavatsalam et al., 2007*; *Riyahi et al., 2011*). Although the catalytic domain of GPCR PIPKs is predicted to have a role in phospholipid signaling, experimental evidence for this hypothesis is still missing. Moreover, it is not known whether G protein signaling is involved downstream of GPCR-PIPKs (*van den Hoogen and Govers, 2018*). Similarly, no ligands for GPCR PIPKs have been identified.

## GPRch3

The GPRch3 cluster contained 28 choanoflagellate GPCRs with short and unstructured N-terminal and C-terminal regions (*Figure 2—figure supplement 4A*); they did not show any similarity to any other previously described GPCR families, either by sequence or structure (*Figure 1B*). Nonetheless, these GPCRs appear to have homologs in diverse metazoans (sponges, ctenophores, cnidarians, protostomes, and deuterostomes), corallochytreans, and non-holozoans (e.g. amoebozoans and alveolates; *Figure 2*). Thus, the GPRch3 GPCRs likely represent a new family of GPCRs with ancient origin in eukaryotes that is retained in metazoans.

## Holozoan GPCRs

### GPR155

GPR155, also known as lysosomal cholesterol sensing (LYCHOS) protein, is an atypical GPCR that binds to lysosomal cholesterol and regulates mTORC1 activity (*Shin et al., 2022*; *Bayly-Jones et al., 2024*; *Schöneberg, 2024*). It contains an unusual 17-transmembrane helix structure composed of an N-terminal transporter-like domain (10 transmembrane helices) fused to a 7TM, with Dishevelled, EGL-10, and Pleckstrin (DEP) domains located at the C terminus. The ability of GPR155 to transduce signals via G proteins is presently unclear. While it has been suggested that the 7TM of GPR155 shares similarities with the aGPCR family (*Bayly-Jones et al., 2024*), it appears to form a cluster of its own in our analysis (*Figure 1B*).

GPR155 has been detected in diverse bilaterians (*Vassilatis et al., 2003*; *Umeda et al., 2017*; *Wang et al., 2018*), and its transporter domain has an ancient origin in eukaryotes (*Dabravolski and Isayenkov, 2022*; *Bayly-Jones et al., 2024*). The finding that GPR155 homologs are encoded in CRMs, but are absent from non-holozoan lineages, suggests that GPR155 originated in the ancestor of holozoans (*Figure 2*). Moreover, the conservation of GPR155 in CRMs extends beyond the 7TM as

both the N-terminal transporter-like domain and the C-terminal DEP domain are detected in GPR155 homologs from CRMs (*Figure 2—figure supplement 5*).

While choanoflagellates and close relatives can produce a variety of sterols (*Kodner et al., 2008*; *Gold et al., 2016*; *Najle et al., 2016*), they are not known to synthesize cholesterol, a vital lipid for metazoans (*Zhang et al., 2019*; *Lebedev et al., 2020*). The ligand(s) of GPR155 in CRMs and whether it mediates sterol-induced signaling await further investigation.

## GPR143

GPR143 homologs were detected in diverse metazoans, including sponges (*Krishnan et al., 2014*), and the filasterean *Capsaspora owczarzaki* (*Figure 2* and *de Mendoza et al., 2014*), but not in choanoflagellates. Therefore, GPR143 likely originated in holozoans and was lost in stem choanoflagellates (*Figure 2*).

While little is known about GPR143, the receptor is expressed in pigment-producing cells in the skin and eyes of metazoans, where it controls melanosome biogenesis, organization, and transport (*Bueschbell et al., 2022*). The downstream signaling pathway is mostly uncharacterized, but GPR143 associates with several Gα and Gβ subunits, along with β-Arrestin (*Schiaffino et al., 1999*; *Schiaffino and Tacchetti, 2005*; *De Filippo et al., 2017*). L-3,4-dihydroxyphenylalanine (L-DOPA) and dopamine have been suggested as ligands for GPR143 (*Lopez et al., 2008*; *Goshima et al., 2019*).

## Stem eukaryote GPCRs that were lost from metazoans
### Rémi-sans-famille (RSF)
28 choanoflagellate receptors clustered together and did not appear to share sequence similarities with other GPCR families previously described in eukaryotes. These GPCR candidates contained a 7TM domain but no additional domains (*Figure 2—figure supplement 6A*). Their predicted structures revealed an additional short helix between TM6 and TM7 with a 'NxLQxxMNxL' conserved motif (*Figure 2—figure supplement 6A and B*). Structural similarity search suggested a weakly supported connection to the THH1/TOM1/TOM3 GPCR family (*Yamanaka et al., 2000*; *Lu et al., 2018*; *Figure 2—figure supplement 6C*).

While these GPCRs are absent from metazoans, we detected homologs in filastereans, as well as in other non-holozoans, including provorans (*Tikhonenkov et al., 2022*), rhodophytes, alveolates, amoebozoans, and ancyromonads (*Figure 2*). Thus, it is likely that this family of GPCRs evolved early during the evolution of eukaryotes and was secondarily lost in various lineages, including metazoans. We named this new class of GPCRs 'Rémi-sans-famille' (RSF), inspired by a fictional orphan (*Malot, 1878*).

### GPRch1
Clustering on the 7TM domains revealed a group of 16 craspedid choanoflagellate GPCRs, which we have named GPRch1 GPCRs. Although these proteins lack a PIPK domain, their 7TM domains resembled those of GPCR PIPKs from choanoflagellates and sponges (*Figure 1B*, *Figure 2—figure supplement 4B*). The GPRch1 7TMs also clustered with 7TMs from amoebozoans, stramenopiles, alveolate, and apusomonadids; interestingly, the non-choanoflagellate GPCRs in this cluster all contained a PIPK domain in the C-terminus. Therefore, we infer that there was a duplication and divergence of GPCR PIPKs in stem eukaryotes, with one of the paralogs giving rise to GPRch1 GPCRs. Subsequently, the choanoflagellate GPRch1 GPCRs lost the PIPK domain. The absence of GPRch1 GPCRs from acanthoecid choanoflagellates, other CRMs, and metazoans suggests that they were lost from these lineages (*Figure 2*).

## Choanoflagellate-specific GPCRs
### GPCR-TKL/K and GPCR-TKs
The GPCR-TKL/K family was identified by examination of a cluster of 176 acanthoecid choanoflagellate GPCRs, 92 of which contain a kinase domain in their C-terminus (*Figures 1 and 2*; *Figure 2—figure supplement 7A–C*). Further analysis showed that the kinase domains of these GPCRs relate to Tyrosine Kinase-like (TKL) domains and other non-tyrosine kinase domains (*Figure 2—figure supplement*

*7B* and Supplementary files 27-29); we therefore named these receptors GPCR-Tyrosine Kinase-Like/Kinase (GPCR-TKL/K).

The fusion between a 7TM and a kinase domain has recently been reported in oomycetes and amoebozoans but was not found in other eukaryotes (*Judelson and Ah-Fong, 2010*; *van den Hoogen et al., 2018*). Thus, the acanthoecid GPCR-TKL/K represents the first observation, to our knowledge, of a GPCR-kinase fusion in holozoans. While the overall protein domain architecture of GPCR-TK/Ks is conserved between choanoflagellates, oomycetes, and amoebozoans, the 7TM domains of acanthoecid GPCR-TKL/K receptors did not cluster with any other GPCRs in eukaryotes (*Figure 2*), suggesting that this family probably originated through convergent evolution in stem acanthoecids (*Figure 2*, *Figure 2—figure supplement 7D*).

Additionally, we detected 11 choanoflagellate GPCRs that contain a predicted tyrosine kinase domain in their C-terminal region, but that did not cluster with the GPCR-TKL/Ks mentioned above (*Figure 2—figure supplement 7B and C*; Supplementary files 27-31). These GPCRs are found in both acanthoecids and craspedids but failed to form a unified family based on their 7TMs (*Figure 2—figure supplement 7C and D*). Unlike their 7TM domains, the kinase domains of these GPCRs clustered together and separately from the GPCR TKL/K kinase domains; all recovered strong blast hits among curated metazoan tyrosine kinases (*Figure 2—figure supplement 7C* and Supplementary file 28). Therefore, we identified these GPCRs as GPCR Tyrosine Kinases (GPCR-TKs).

While no metazoan homologs were found when using the 7TM domain of choanoflagellate GPCR-TKs as queries, using the conserved tyrosine kinase domains as queries recovered GPCR-TKs in sponges but not in other metazoan lineages or other holozoans (*Figure 2—figure supplement 7E*). To test whether GPCR-TKs in sponges and choanoflagellates are homologous, we performed phylogenetic analyses of their TK and 7TM domains (*Figure 2—figure supplement 7F and G*; Supplementary files 32-35). While the TK domains of GPCR-TKs from sponges and choanoflagellates formed a well-supported clade, their 7TM domains did not. These results point to a heterogeneous evolutionary history that may include domain swapping (i.e. ancestral GPCR-TKs in which the 7TM domain was replaced in either the sponge or choanoflagellate lineages) or convergent evolution, in which homologous 7TM domains fused with unrelated 7TM domains in the sponge and choanoflagellate lineages.

Together, our data suggest that the fusion of a C-terminal kinase domain to a 7TM domain likely evolved repeatedly and independently among eukaryotes and within the choanoflagellate phylogeny. The evolution of GPCR-TKs seems to be restricted to choanoflagellates and early-branching metazoans. The evolutionary and molecular implications underlying the fusion of a 7TM domain with an intracellular kinase domain are unknown, and, to our knowledge, no GPCR-TK, GPCR-TKL, or GPCR-Ks have been functionally characterized.

## Additional GPCRs
### GPRch2
A cluster of 12 choanoflagellate GPCRs showed no similarity to previously identified GPCR families (*Figure 1*). Although these GPCRs do not possess additional protein domains outside their signature 7TM domain (*Figure 2—figure supplement 8A*), the GPRch2 GPCRs exhibit an unusual large intracellular loop 3 (ICL3) linking the TM5 to the TM6 (*Figure 2—figure supplement 8A and B*).

While homologs of these GPCRs were also found in filastereans, ichthyosporeans, and corallochytreans, they are absent from metazoans and were only detected in green algae outside of holozoans (*Figure 2*). Thus, it is likely that the GPRch2s evolved in stem holozoans either de novo or via a horizontal transfer from green algae and were subsequently lost in stem metazoans.

### Frizzled
The Frizzled family, which encompasses Frizzled GPCRs and Smoothened, controls cell proliferation, cell fate, tissue polarity, and cell polarity during metazoan development (*Taipale and Beachy, 2001*; *Logan and Nusse, 2004*; *Schulte, 2024*). These receptors have a highly conserved cysteine-rich domain called the Frizzled (Fz) domain, located in their extracellular N-terminus (*Lagerström and Schiöth, 2008*; *Zheng and Sheng, 2024*). This domain is connected to the 7TM region by a linker segment. The Fz domain serves as the ligand-binding site for most Frizzled receptors. Several ligands

have been found to activate members of the Frizzled family, of which the secreted glycoprotein Wingless/Int-1 (Wnt) has been the most studied (**Schulte, 2024**).

Although initially characterized in *Drosophila*, Frizzled receptors have been detected in most metazoans, including sponges (**Figure 2**; **Krishnan et al., 2014**; **Holzem et al., 2024**). While Frizzled receptors have not been previously detected in CRMs, homologs were found in fungi and amoebozoans, suggesting a probable origin of the family in the last common ancestor of opisthokonts and amoebozoans (**Krishnan et al., 2012**; **de Mendoza et al., 2014**). Notably, fungal and amoebozoan Frizzled homologs also possess a Fz domain in their N-termini, supporting an overall conserved receptor topology.

While we could not detect members of the Frizzled family in any of the 23 choanoflagellate species used in our study (**Figure 2**), we found four Frizzled/Smoothened receptors in the transcriptome of the corallochytrean *Syssomonas multiformis* (**Figure 2**; **Figure 2—figure supplement 9A and B**; Supplementary file 36). However, contaminating sequences have been reported in this data set, and therefore, this observation should be treated with caution (**Hehenberger et al., 2017**).

## Discussion

Our transcriptomic and genomic survey demonstrates that choanoflagellate GPCRomes are richer than previously understood, both in the number and diversity of GPCRs represented. We identified 18 distinct GPCR families in choanoflagellates, of which 12– Rhodopsin, GPR155, GPR157, GPCR TKL/K, TMEM145, TMEM87, GPR180, Hi-GOLD, RSF, GPRch1, GPRch2, and GPRch3– were newly described in these organisms (**Table 1**, **Figure 2**). Among these families, we observed that aGPCRs generally constitute the largest class of GPCRs in choanoflagellates, apart from the GPCR TKL/K family in acanthoecids. Most choanoflagellate GPCR families are conserved in metazoans or other eukaryotes. This supports the view that GPCRs are ancient gene families found across eukaryotic diversity (**Krishnan et al., 2012**; **de Mendoza et al., 2014**; **Mojib and Kubanek, 2020**).

In addition, by assessing the conservation of choanoflagellate GPCRs in other eukaryotes, we uncovered GPCR families in metazoans that, to our knowledge, had not been reported in this clade before. These include Hi-GOLD, GPRch3, and GPCR TKL/Ks. In addition, we found that five GPCR families– RSF, Hi-GOLD, GPRch3, GPR157, and GPRch1– are conserved across diverse eukaryotes, thereby expanding the repertoire of GPCRs inferred at the root of the eukaryotic tree (**de Mendoza et al., 2014**). Future increases in the sequencing of non-metazoans will likely expand further the inferred diversity of GPCRs present in the progenitors of metazoans and other eukaryotes.

The identification of Glutamate Receptors, Rhodopsins, and aGPCRs in choanoflagellates, three GPCR families with relatively well-characterized subfamilies in metazoans (**Fredriksson et al., 2003**; **Pin et al., 2003**; **Bjarnadóttir et al., 2004**; **Chun et al., 2012**; **Scholz et al., 2019**; **Ellaithy et al., 2020**; **Wittlake et al., 2021**), led us to investigate whether these subfamilies are conserved in choanoflagellates and close relatives. Based on analyses of the 7TM domains, we found relatives of the main subfamilies of metazoan Glutamate Receptors in CRMs (**Figure 3B**). In contrast, most aGPCR subfamilies reported in metazoans have no obvious orthologs in CRMs, suggesting that metazoan and CRM aGPCRs diversified independently (**Figure 5**). One exception was the ADGRV family (**Weston et al., 2004**; **Hamann et al., 2015**; **Kusuluri et al., 2021**), for which clear orthologs are found in choanoflagellates and filastereans (this study and **Krishnan et al., 2012**; **Peña et al., 2016**).

Although Rhodopsins are the most abundant GPCRs in metazoans, we identified only three metazoan-like Rhodopsins in CRMs: two in choanoflagellates and one in an ichthyosporean. In addition, we uncovered Rhodopsin-like GPCRs from diverse fungi and amoebozoans (Supplementary file 9), corroborating the findings of prior studies (**Krishnan et al., 2012**; **de Mendoza et al., 2014**). Sequence comparisons and structural predictions suggest that the two choanoflagellate Rhodopsins most closely resemble metazoan opsins, a Rhodopsin subfamily (**Fredriksson et al., 2003**; **Mickael et al., 2016**). Opsins, light-detecting receptors supporting vision, have only been described in metazoans (**Feuda et al., 2012**; **Fleming et al., 2020**; **Wong et al., 2022**; **Hagen et al., 2023**; **Aleotti et al., 2025**). Unlike metazoan opsins, the two choanoflagellate Rhodopsins lack canonical residues essential for light-sensing, including the highly conserved Lysine K296, which mediates binding to the chromophore retinal (**Davies et al., 2010**; **Hankins et al., 2014**; **Nagata and Inoue, 2021**). Thus, while it is formally possible that Rhodopsins existed in stem choanoflagellates and were lost in most modern choanoflagellate lineages, either horizontal gene

transfer or convergent evolution in the shared ancestor of *S. macrocollata* and *S. punica* are similarly plausible explanations for their presence in these species. Differentiating between these alternative evolutionary scenarios is challenging because of the rapid rate of sequence evolution within the family and the resultant loss of phylogenetic signal. Our own preliminary investigations of Rhodopsin evolution in non-metazoans were inconclusive. Therefore, ambiguities about the provenance and function of CRM Rhodopsins currently obscure the ancestry of metazoan Rhodopsins and opsins.

The N-terminal and C-terminal protein domains of choanoflagellate, other CRM, metazoan, and additional eukaryotic GPCRs revealed uneven evolutionary conservation between GPCR families. For example, we found that the C-terminal PIPK domain of the GPCR PIPK family, the N-terminal transporter domain and C-terminal DEP domain of GPR155, and the N-terminal GOLD domain of GOST GPCRs are generally conserved in CRMs, metazoans, and other eukaryotes. In contrast, the stereotypical association of N-terminal protein domains from metazoan Glutamate Receptors is absent from non-metazoan Glutamate Receptors identified by 7TM clustering. Moreover, we observed that the N-termini of Glutamate Receptors from filastereans and holomycotans appear to possess a larger diversity of protein domains than metazoan Glutamate Receptors, suggesting functional constraint exerted on the architecture of metazoan Glutamate Receptors. This likely has implications for both the ligands that non-metazoan Glutamate Receptors may recognize and their underlying mechanism of activation (*Hu et al., 2000*; *Jiang et al., 2004*; *Rondard et al., 2006*; *Levitz et al., 2016*; *Ellaithy et al., 2020*; *Laboute et al., 2023*).

The aGPCR family shows the greatest level of protein domain composition diversity in metazoans (*Lagerström and Schiöth, 2008*; *Krishnan et al., 2012*; *Hamann et al., 2015*). Similarly, we uncovered a large diversity of N-terminal protein domain combinations in CRM aGPCRs, with a clear diversification of the repertoire in choanoflagellates and, to a lesser extent, in filastereans (*Figures 1B and 5*, *Figure 5—figure supplement 2*). The aGPCRs in CRMs displayed all the core structural hallmarks of metazoan aGPCRs, including a long N-terminus containing a GAIN domain and many additional ECDs. Moreover, we found that the HRM/GAIN/7TM module, previously thought to be restricted to metazoans, is also found in CRM aGPCRs. Our data support an evolutionary scenario wherein the linkage between the GAIN and 7TM domains evolved first in stem holozoans, followed by the incorporation of the HRM into the HRM/GAIN/7TM module in the progenitors of Filasterea and Choanozoa.

Despite its conservation, the function of the HRM-GAIN-7TM module is presently unclear (*Shima et al., 2004*; *Kimura et al., 2006*; *Araç et al., 2012*; *Prömel et al., 2012*; *Araç and Leon, 2019*). Interestingly, the HRM domain binds peptide hormones and neuropeptides in the Secretin GPCR family (*Lagerström and Schiöth, 2008*; *Pal et al., 2012*; *Zhao et al., 2023*). The presence of HRM-containing aGPCRs in CRMs could suggest a pre-metazoan origin of peptide-based endocrine signaling. This is reinforced by the recent finding that sponges and choanoflagellates express sequences that resemble metazoan neuropeptides and peptide hormones (*Steffen et al., 2021*; *Yanez-Guerra et al., 2022*; *Lin et al., 2024*).

Finally, the diversity of protein domain architectures in the aGPCRs of choanoflagellates and metazoans suggests that aGPCR evolution was shaped by extensive domain shuffling (*Gilbert, 1978*; *Patthy, 1999*; *Smithers et al., 2019*; *Patthy, 2021*). While some discrete protein domain associations (GAIN/7TM and HRM/GAIN/TM) or, in rare cases, most of the receptor sequence (ADGRV/subfamily VIII) survived, presumably under selection, new protein domain architectures arose, perhaps through a combination of gene duplication followed by domain shuffling. The diversification of NTF domains might have facilitated the recognition of a diversity of ligands (e.g. proteoglycans and proteins).

Of course, the aGPCRs were named for the presence of adhesion-related protein domains in the NTFs of metazoan aGPCRs, some of which support cell-cell and cell-matrix adhesion (*Langenhan et al., 2013*; *Hamann et al., 2015*). The relevance of adhesion domains in CRM aGPCRs is intriguing and requires further investigation. Interestingly, diverse CRMs form facultative multicellular structures (*Fairclough et al., 2010*; *Marshall and Berbee, 2011*; *Glockling et al., 2013*; *Sebé-Pedrós et al., 2013*; *Carr et al., 2017*; *Brunet et al., 2019*; *Dudin et al., 2019*; *Tikhonenkov et al., 2020*; *RosRocher et al., 2021*). Therefore, the diversification of adhesion-related protein domains in the NTFs of aGPCRs may have contributed to the evolution of multicellularity in holozoans. In addition, a recent study in sponges suggested that aGPCRs could be part of the immune response following exposure to microbial-associated molecular patterns (*Pita et al., 2018*).

A better appreciation of the GPCRs expressed in choanoflagellates and other CRMs is a first step to understanding cue sensing in these organisms. While environmental stimuli are known to play an essential role in controlling life history transitions in choanoflagellates, the receptors involved have yet to be characterized (*Dayel et al., 2011*; *Alegado et al., 2012*; *Levin and King, 2013*; *Woznica et al., 2017*; *RosRocher and Brunet, 2023*). We anticipate that the identification of GPCR repertoires in the sister group of metazoans will open the way to the functional characterization of this superfamily of receptors in choanoflagellates, which could illuminate the origin and ancestral function(s) of key signaling pathways in metazoans.

## Materials and methods
### Taxon sampling
The taxa surveyed in this study are listed in *Table 2*.

**Table 2.** Taxon sampling and data sources for genomes and transcriptomes analyzed in this study.

| Group | Species | Source |
|---|---|---|
| Choanoflagellate proteomes | *Diaphanoeca grandis, Acanthoeca spectabilis, Salpingoeca punica, Salpingoeca urceolata, Hartaetosiga gracilis, Salpingoeca macrocollata, Stephanoeca diplocostata, Helgoeca nana, Salpingoeca kvevrii, Didymoeca costata, Choanoeca perplexa, Salpingoeca infusionum, Microstomoeca roanoka, Savillea parva, Hartaetosiga balthica, Salpingoeca dolichothecata, Codosiga hollandica, Salpingoeca helianthica, Mylnosiga fluctuans, Salpingoeca rosetta, Monosiga brevicollis, Barroeca monosierra, Choanoeca flexa* | NCBI BioProject PRJNA419411, Figshare DOI: 10.6084 /m9.figshare.5686984.v2 *Richter et al., 2018*, NCBI BioProject PRJNA37927 *Fairclough et al., 2013* ,GenBank GCA_000002865.1 *King et al., 2008*, Figshare DOI: 10.6084 /m9.figshare.8216291 *Brunet et al., 2019*, *Barroeca monosierra* genome-derived proteome *Hake et al., 2024*, EukProt: https://evocellbio.com/eukprot/ |
| Filasterean proteomes | *Pigoraptor vietnamica, Pigoraptor chileana, Capsaspora owczarzaki, Ministeria vibrans* | MulticellGenome: https://multicellgenome.com/genomes |
| Ichthyosporean proteomes | *Sphaeroforma arctica, Creolimax fragrantissima, Ichthyophonus hoferi, Abeoforma whisleri, Pirum gemmata, Chromosphaera perkinsii* | MulticellGenome: https://multicellgenome.com/genomes |
| Corallochytrean proteomes | *Syssomonas multiformis, Corallochytrium limacisporum* | MulticellGenome: https://multicellgenome.com/genomes; https://doi.org/10.5061/dryad.26bv4 |
| Metazoan proteomes | *Trichoplax adhaerens, Amphimedon queenslandica, Corticium candelabrum, Dysidea avara, Ephydatia muelleri, Halichondria panicea, Oscarella lobularis, Oopsacas minuta, Tethya wilhelma, Mnemiopsis leidyi, Bolinopsis microptera, Danio rerio, Gallus gallus, Drosophila melanogaster, Ciona intestinalis, Branchiostoma floridae,* | UniProt Proteomes: https://www.uniprot.org/ |
| | *Nematostella vectensis, Acropora millepora, Caenorhabditis elegans, Mus musculus, Homo sapiens* | |
| Holomycota proteomes | *Paraphelidium tribonemae, Neurospora crassa, Emericella nidulans, Ustilago maydis, Yarrowia lipolytica, Cryptococcus neoformans, Saccharomyces cerevisiae, Schizosaccharomyces pombe, Fonticula alba, Rozella allomycis, Allomyces macrogynus, Conidiobolus coronatus, Batrachochytrium dendrobatidis, Rhizophagus irregularis, Rhizopus delemar* | UniProt Proteomes https://www.uniprot.org/ |
| Amoebozoan proteomes | *Dictyostelium purpureum, Heterostelium pallidum, Entamoeba histolytica, Dictyostelium discoideum, Cavenderia fasciculata, Acanthamoeba castellanii, Planoprotostelium fungivorum, Polysphondylium violaceum, Dictyostelium firmibasis* | UniProt Proteomes: https://www.uniprot.org/ |
| Viridiplantae proteomes | *Volvox carteri, Chlamydomonas reinhardtii, Chlorella variabilis, Ostreococcus tauri, Gonium pectorale, Bathycoccus prasinos, Chlamydomonas eustigma, Raphidocelis subcapitata, Chloropicon primus, Chlamydomonas schloesseri, Pycnococcus provasolii, Volvox reticuliferus, Coccomyxa sp., Pleodorina starrii, Arabidopsis thaliana, Ricinus communis, Oryza sativa, Rosa chinensis, Vigna angularis, Rhododendron simsii, Papaver atlanticum, Cryptomeria japonica* | UniProt Proteomes: https://www.uniprot.org/ |
| Ciliate proteomes | *Paramecium tetraurelia, Ichthyophthirius multifiliis, Tetrahymena thermophila, Stylonychia lemnae, Pseudocohnilembus persalinus, Stentor coeruleus, Halteria grandinella, Euplotes crassus* | UniProt Proteomes: https://www.uniprot.org/ |

## Identification of choanoflagellate GPCRs

### GPCR mining through sequence homology-based approaches

To identify putative choanoflagellate GPCR sequences in the 23 choanoflagellate species (*Table 2* and Supplementary file 1), the choanoflagellate predicted proteomes were searched with HMMs unique to each of the 54 GPCR families belonging to the GPCR_A Pfam clan (CL0192; Supplementary file 2; *Mistry et al., 2021*) using the HMMER3.3 software package (cutoff threshold value: 0.01; *Eddy, 2011*). We recovered 1070 candidate GPCRs. In a complementary approach, we used a global HMM profile (GPCRHMM, local score) that mimics the common topology of GPCRs (*Wistrand et al., 2006*) and recovered 1095 candidates in our analysis. The comparison of the two sets of GPCRs revealed 381 duplicate candidates, which were merged, leaving a final set of 1784 candidates.

### Validation of candidate GPCRs through domain composition and topology filtering

To reduce false positives and remove highly truncated sequences and possible isoforms, we submitted the candidate GPCRs to a round of filtering that combined complementary approaches. All the candidates were used as queries to perform BLASTP searches against both the NCBI and GPCRdb databases (*Altschul et al., 1990*; *Pándy-Szekeres et al., 2023*) using an e-value cut-off of $1\times10^{-5}$. The sequences that recovered a GPCR within the 10 top hits were kept. In addition, GPCR candidates were evaluated on the basis of their protein domain composition by combining the search tools Inter-ProScan, CDvist, and the transmembrane domain predictor TMHMM 2.0 (*Krogh et al., 2001*; *Adebali et al., 2015*; *Blum et al., 2025*). The prediction of a 7TM domain, along with additional signature protein domains, was assessed for each candidate. Protein sequences derived from genomes and transcriptomes were retained if containing six to eight transmembrane helices to account for the detection of possible N-terminal signal peptides (*Rutz et al., 2015*) and TMHMM prediction errors. Sequences that possessed a 7TM domain but that did recover GPCRs when used as BLAST queries were subjected to a comparative structural analysis by predicting their structure using AlphaFold 3 (*Abramson et al., 2024*) and searching for structural homologs within the AlphaFold databases (*van Kempen et al., 2024*) see 'Protein structure prediction and search' below for more details on the analysis. Candidates with a predicted barrel shape conformation of their 7 alpha helices and that recovered GPCR structural hits (E-value<$1e^{-9}$) were considered GPCRs and were kept in our dataset. Finally, we removed putative splice isoforms using CD-HIT with a 90% identity threshold (*Li and Godzik, 2006*). A total of 1113 false positives, isoforms, and highly truncated sequences were identified and removed from our dataset, leaving 671 validated choanoflagellate GPCRs in total.

### Recovering additional choanoflagellate GPCRs using choanoflagellate GPCR BLAST queries and custom choanoflagellate GPCR HMMs

Because the HMMs used to search for choanoflagellate GPCRs in our analysis were primarily based on seed alignments of metazoan sequences, we looked for additional choanoflagellate GPCRs by using the 671 validated choanoflagellate GPCRs as queries. To this end, we simplified the choanoflagellate GPCRs dataset by clustering the 671 choanoflagellate GPCRs based on all-against-all pairwise sequence similarities (see 'Clustering of the 918 validated choanoflagellate GPCRs' below to find more details about the protocol). The 671 choanoflagellate GPCRs were sorted into 18 clusters, with only 76 GPCRs that did not group with other GPCRs.

Next, we built cluster-specific GPCR HMMs for each of the 18 choanoflagellate GPCR clusters previously identified. We also built individual HMMs for the 76 GPCRs that did not fall into the 18 clusters. To this end, the 7TM domains of all the members of each of the 18 GPCR clusters were aligned independently using MAFFT (Version 7.4) with the E-INS-i algorithm (*Katoh et al., 2018*). The resulting 18 multisequence alignments were then used to build 18 corresponding choanoflagellate cluster-specific GPCR HMMs using the hmmbuild module of HMMER v3.3. In the case of GPCRs that did not belong to any clusters, HMMs were built based on single 7TM domains. These HMMs were used to search the choanoflagellate proteomes again. In parallel to this approach, we also selected three choanoflagellate GPCR sequences per GPCR cluster or a single GPCR sequence in the case of isolated GPCRs, to use as BLASTP queries against the 23 choanoflagellate proteomes (E value: $1.0e^{-5}$). The resulting GPCR candidates were validated through the filtering approach previously described,

leaving 247 new GPCRs that were not predicted with the metazoan-biased HMMs. Added to the original 671, this yielded a total of 918 GPCRs from the 23 choanoflagellate species. The complete list of validated choanoflagellate sequences in FASTA format and the choanoflagellate-specific GPCR HMMs are available in Supplementary files 3 and 4.

## Identification of GPCR signaling pathway components

Heterotrimeric G proteins (Gα, Gβ, and Gγ), positive regulators (Phosducin and Ric8), and negative regulators (Arrestin, GRK, RGS, and GoLoco) of G protein signaling were identified by searching choanoflagellate proteomes with Pfam HMM profiles using the HMMER3.3 software package (*Eddy, 2011*; cutoff threshold value: 1e$^{-5}$) with the hmmsearch function. The following Pfam HMMs were used: PF00503 (G protein alpha subunit), PF00400 (WD domain, G-beta repeat), PF00631 (GGL domain), PF02752 (Arrestin (or S-antigen), C-terminal domain), PF00339 (Arrestin (or S-antigen), N-terminal domain), PF02188 (GoLoco motif), PF02114 (Phosducin), PF00615 (Regulator of G protein signaling domain), and PF10165 (Ric8; Supplementary file 2). The complete protein domain architecture of the candidates was assessed with CDvist (*Adebali et al., 2015*) and choanoflagellate sequences from each category were then used as BLAST queries against the 23 choanoflagellate proteomes on the EukProt database (E value: 1.0e$^{-5}$; *Richter et al., 2022*) to look for additional sequences that might not have been detected in the previous round of screening. All the sequences identified in this analysis are listed in Supplementary file 5.

## Identification of HRM-containing proteins

To identify HRM-containing proteins across eukaryotes, we searched the proteomes of metazoans, choanoflagellates, filastereans, ichthyosporeans, corallochytreans, holomycotans, amoebozoans, viridiplantae, and ciliates with the Pfam HMM profile HRM (PF02793) using the HMMER3.3 software package (cutoff threshold value: 1e$^{-5}$) with the hmmsearch function (*Eddy, 2011*). The complete protein domain architecture of the candidates was then further assessed using Cdvis (*Adebali et al., 2015*). The five HRM-containing proteins identified in the nucleariid *F. alba* are provided in Supplementary file 24.

## Identification of GPCR families in other eukaryotes

To assess the presence of Glutamate, Rhodopsin, Adhesion, and Frizzled GPCR families in other eukaryotes, we searched the relevant genomes and transcriptomes (*Table 2*) with their signature 7TM HMMs (Supplementary file 2) using the HMMER3.3 software package (cutoff threshold value: 0.01) with the hmmsearch function. The candidates were then validated through the previously described domain composition and topology filtering pipeline (see 'Validation of candidate GPCRs through domain composition and topology filtering') (Supplementary file 9).

To assess the presence of other GPCR families, we searched the entire Eukprot v3 dataset (993 species) with previously published metazoan and/or non-metazoan GPCR family members as queries (threshold E value: 1.0e$^{-5}$). Both the full-length GPCR and the extracted 7TM domains were used as search queries in this analysis. We defined hits as those with at least 70% query coverage and at least 30% sequence identity. When no hits were recovered, an additional search using their signature 7TM HMMs (Supplementary file 2) was performed on the genomes/transcriptomes dataset (*Table 2*). All the candidates were then validated through a similar domain composition and topology filtering approach.

## Clustering analyses
### Clustering of the 918 validated choanoflagellate GPCRs

To sort the choanoflagellate GPCRs into families, we created a dataset composed of all the 7TM regions extracted from the 918 choanoflagellate sequences identified in our study, along with the 7TM sequences of various metazoan, amoebozoan, stramenopile, and chlorophyte GPCRs to aid with the identification of the clusters (Supplementary file 8). The non-choanoflagellate sequences added to the dataset were either top blast hits recovered after searching the entire Eukprot v3 dataset (993 species) with choanoflagellate GPCRs as queries, or previously published and well-documented GPCR sequences from metazoans. To isolate the 7TM region, we aligned all the GPCR sequences on

Geneious Prime v2024.07 using MAFFT with E-INS-I algorithm (*Katoh et al., 2018*), and predicted their transmembrane helices with TMHMM. All the sequences starting from the N-terminal end of the transmembrane helix 1 and finishing at the C-terminal end of the transmembrane helix 7 were kept.

Next, due to the inherent difficulties of analyzing a large set of proteins with mild sequence conservation without decreasing the accuracy of the alignment used for phylogenetic inference, we opted for an all-against-all pairwise similarity-based clustering approach to assess the diversity of GPCR families in choanoflagellates using the clustering tool CLANS 2.2.2 (*Frickey and Lupas, 2004*; *Gabler et al., 2020*). Unlike phylogenetic reconstruction, this approach becomes more accurate with an increasing number of sequences (*Frickey and Lupas, 2004*). First, we used the CLANS web utility in the MPI Bioinformatic toolkit (https://toolkit.tuebingen.mpg.de/tools/clans) to perform an all-against-all BLAST search (with scoring matrix BLOSUM62) to obtain a matrix of pairwise sequence similarities with a cut-off value of $1e^{-6}$ for BLAST E-values (High Scoring Pairs with E-value higher than $1e^{-6}$ were not extracted). We then opened the resulting matrix in the CLANS graphical interface, and CLANS was allowed to cycle 20,000 times to optimize the graph. Clusters were automatically detected using the convex clusters search of four or more proteins with the attraction value limit set at 0.5 standard deviation. A total of 18 choanoflagellate GPCR clusters were recovered. The clustering maps were further edited in Adobe Illustrator 2024 to modify the colors and symbols used.

## Clustering of holozoan aGPCR N-termini

The extracellular region of the metazoan, choanoflagellate, and filasterean aGPCRs was extracted by combining the transmembrane domain prediction tool TMHMM-2.0 (*Krogh et al., 2001*) with the alignment tool MAFFT (E-INS-I algorithm) on Geneious Prime. All the regions starting at the N-terminal end of the GPCRs and finishing at the start of the first transmembrane helix were retained for downstream analysis (Supplementary file 21). The clustering analysis performed was comparable to the analysis described previously for GPCRs, with a $1e^{-20}$ cut-off value being used for the matrix pairwise sequence similarities.

## Clustering of rhodopsins

To compare the newly identified choanoflagellate Rhodopsins with the Rhodopsins found in a diversity of metazoans and other CRMs, we searched the proteomes of metazoans and CRMs (*Table 2*) with the Pfam HMM profile 7TM_1/Rhodopsin (PF00001) using the HMMER3.3 software package (cutoff threshold value: 1e-5) with the hmmsearch function. Candidates were then validated by assessing their InterProScan protein domain signature and BLASTing the sequences against the NCBI database. Additionally, we used the two choanoflagellate Rhodopsins – *S.macrocollata*_m.143379 and *Salpingoeca_punica*_m.44256 – to query the EukProt database (993 species), and the top blast hits with E values $<1 \times 10^{-10}$ were included in our dataset. A total of 6149 validated Rhodopsins were then submitted for all-against-all pairwise comparison with a $1e^{-12}$ cut-off value being used for the matrix pairwise sequence similarities. All the validated Rhodopsins used in the analysis and the output of the analysis are found in Supplementary files 13 and 14.

## Clustering of GPCR Kinase and kinase domains

To assess the diversity of choanoflagellate GPCR Kinase along with the diversity of their kinase domains, we extracted their 7TM domain and their kinase domain separately. To this end, we aligned all the choanoflagellate GPCR Kinase sequences on Geneious Prime using the alignment tool MAFFT (E-INS-I algorithm). The 7TM domains and the kinase domains were then predicted and extracted by running locally TMHMM-2.0 and Interproscan (PfamA; Supplementary files 27-30). The clustering analyses were similar to the ones described previously, with cut-off values of $1e^{-6}$ and $1e^{-20}$ for the 7TM and kinase domains, respectively. In parallel, each kinase domain was also blasted against KinBase (*Manning et al., 2002*; *Bradham et al., 2006*; *Goldberg et al., 2006*), the curated protein kinase dataset from https://www.kinase.com, for identification (Supplementary file 29).

## Sequence alignment and phylogenetic analyses

Phylogenetic analyses of aGPCRs, Glutamate Receptors, Gα subunits, and the 7TM and Kinase domains from GPCR TK/TKL/Ks were performed in this study. To construct the holozoan aGPCR and Glutamate

Receptor phylogenies, we first extracted the aGPCR and Glutamate Receptor 7TM domains by combining the transmembrane domain prediction tool TMHMM-2.0 with the alignment tool MAFFT (E-INS-I algorithm) on Geneious Prime v2024.07; only the protein sequences spanning the 7TM region were retained for downstream analysis (Supplementary files 10, 17 and 19). To reduce the size of the large GPCR datasets while preserving their diversity, we systematically clustered similar sequences using CD-HIT (threshold 0.8; word size = 5). A total of 257 Glutamate Receptor 7TM sequences and 659 aGPCR 7TM sequences were kept for downstream analyses. To construct the phylogenies of the Kinase domain and 7TM domain from the GPCR TK/TKL/Ks, we first built a dataset including all the GPCR TK/TKL/Ks sequences identified in choanoflagellates and in sponges, as well as the GPCR TKL/Ks previously published in oomycetes and amoebozoans (*van den Hoogen et al., 2018*). We extracted the 7TM domain and Kinase domain from each sequence by combining the transmembrane domain prediction tool TMHMM-2.0 and the protein domain prediction tool InterProScan with the alignment tool MAFFT (E-INS-I algorithm) on Geneious Prime v2024.07 (Supplementary files 32 and 34). We then aligned the aGPCR, Glutamate and GPCR TK/TKL/K Receptor 7TMs, the GPCR TK/TKL/Ks Kinase domain, or the full-length Gα sequences using MAFFT with the E-INS-I algorithm. The resulting alignments were then used for Maximum-likelihood and/or Bayesian inference of phylogenies (*Figures 3B and 5A*, *Figure 1—figure supplement 1D*, *Figure 5—figure supplement 1A*, and *Figure 2—figure supplement 7F and G*; Supplementary files 7, 11, 18, 20, 33, and 35). We built Maximum-likelihood phylogenies with IQ-TREE web server (http://iqtree.cibiv.univie.ac.at/; *Trifinopoulos et al., 2016*) using ModelFinder (*Kalyaanamoorthy et al., 2017*) and 2000 Ultrafast Bootstraps (UF-boot) (*Minh et al., 2013*) or 2000 iterations of SH-aLRT. Bayesian inference was performed using MrBayes v3.2.7a (*Huelsenbeck and Ronquist, 2001*; *Ronquist et al., 2012*) on CIPRES Science Gateway (*Miller et al., 2010*). The posterior probability of the trees was estimated using Markov Chain Monte Carlo (MCMC) analysis and a fixed LG amino acid substitution model (Aamodelpr = Fixed(LG)). The gamma-shaped model was used to estimate the variation of evolutionary rates across sites (set rates = Gamma). MCMC analysis was set to run for 6,000,000 generations and every 120th tree was sampled. Diagnostics were calculated for every 1000 generations (diagnfreq = 1000) to analyze the convergence of the two independent runs starting from different random trees. To terminate the MCMC generations, a stop rule was applied, and the convergence was analyzed until the average standard deviation of split frequencies dropped below 0.01. To ensure that the parameter estimates were only made from data drawn from distributions derived after the MCMCs had converged, we discarded the first 25% of the sampled trees in the burn-in phase using relative burn-in setting (relburnin = yes and burninfrac = 0.25). A consensus tree was built from the remaining 75% of the sampled trees with the MrBayes sumt command using the 50% majority rule method. Trees were visualized using iTOL (https://itol.embl.de/; *Letunic and Bork, 2024*) and further edited in Adobe Illustrator 2024.

## Protein domain search

To identify and compare the protein domains found in the extracellular region of aGPCRs and Glutamate Receptors across a range of eukaryotes, we first assessed the aGPCR and Glutamate Receptor repertoires from various metazoans, filastereans, ichthyosporeans, corallochytreans, holomycotans, amoebozoans, viridiplantae, and ciliates (see *Table 2*). Briefly, the proteomes were searched for aGPCR and Glutamate Receptors with the Pfam HMM profiles – 7TM_2/Adhesion (PF00002) and 7TM_3/Glutamate (PF00003) – respectively, using the HMMER3.3 software package (cutoff threshold value: $1e^{-5}$) with the hmmsearch function. Candidates were further verified by assessing their InterProScan protein domain signature and BLASTing the sequences against the NCBI database. The validated aGPCR and Glutamate Receptors from each clade (choanoflagellates, metazoans, filastereans, ichthyosporeans, corallochytreans, holomycotans, amoebozoas, viridiplantae, and ciliates) were then analyzed in batch using CDvist with default parameters to assess their protein domain composition (*Figure 3C*, *Figure 5—figure supplement 2*; Supplementary files 12 and 21).

## Protein structure prediction and search

The protein structures of full-length GPCRs and/or extracted 7TM domains were predicted using the AlphaFold 3 server with default parameters (https://alphafoldserver.com/; *Abramson et al., 2024*). Five models were generated per input, of which only the top-ranked prediction (based on the ranking_score metric) was selected for downstream analyses. The quality of the predicted structure

was then assessed based on the pLDDT confidence score. Only models with pLDDT scores above 70 were retained. Structures were analyzed and figures were prepared with ChimeraX (*Meng et al., 2023*).

To find structural homologs, we searched the AlphaFold-predicted structures against the AFDB-PROTEOME, AFDB-SWISSPROT, and AFDB50 databases using the search program Foldseek (https://search.foldseek.com/search; *van Kempen et al., 2024*) with default parameters. Structural top hits with a pLDDT confidence score above 70 in the aligned region and an associated E-value<$1e^{-9}$ were considered putative structural homologs.

### Logo analysis

To build GAIN domain sequence logos (*Figure 5—figure supplement 4B*), we isolated the C-terminal region of the GAIN domain from 301 choanoflagellate, 99 filasterean, 1 corallochytrean, and 30 murine aGPCRs (Supplementary file 23). To do so, we aligned the aGPCR sequences from the four datasets separately using MUSCLE v5 (*Edgar, 2022*) on Geneious Prime, and the region of the C-terminal GAIN domain starting with the first N-terminal conserved Cysteine and finishing with the TA consensus 'TxFAVLM' was extracted and kept for downstream analysis. We then trimmed the aligned sequences with ClipKIT (*Steenwyk et al., 2020*) using the GAPPY mode (with the default value:0.9). The resulting processed alignments were analyzed with WebLogo3 (https://weblogo.threeplusone.com/; *Crooks et al., 2004*).

## Acknowledgements

We are grateful to Demet Araç, Thibaut Brunet, Daniel Richter, and Iñaki Ruiz-Trillo for feedback on the manuscript. We thank the whole King lab for valuable feedback throughout the project and Flora Rutaganira for advice about categorizing kinases. This work was supported by a Human Frontier Science Program long-term fellowship (LT 000919/2020 L) to AGDLB and an Investigator Award from the Howard Hughes Medical Institute (NK).

## Additional information

### Funding

| Funder | Grant reference number | Author |
| --- | --- | --- |
| Human Frontier Science Program | LT 000919/2020-L | Alain Garcia De Las Bayonas |
| Howard Hughes Medical Institute | | Nicole King |

The funders had no role in study design, data collection and interpretation, or the decision to submit the work for publication.

### Author contributions

Alain Garcia De Las Bayonas, Conceptualization, Data curation, Formal analysis, Investigation, Methodology, Writing – original draft, Writing – review and editing; Nicole King, Conceptualization, Supervision, Funding acquisition, Writing – original draft, Project administration, Writing – review and editing

### Author ORCIDs

Alain Garcia De Las Bayonas https://orcid.org/0000-0002-2415-7147
Nicole King https://orcid.org/0000-0002-6409-1111

Reviewer #1 (Public review): https://doi.org/10.7554/eLife.107467.4.sa1
Reviewer #2 (Public review): https://doi.org/10.7554/eLife.107467.4.sa2
Author response https://doi.org/10.7554/eLife.107467.4.sa3

# Additional files

**Supplementary files**
MDAR checklist

## Data availability

Our manuscript is a computational study; no new sequencing data have been generated for this manuscript. All the raw data analyzed in this study are publicly available (see *Table 2* under the '"Materials and methods'" section and Supplementary File 1). All the supporting raw data and results of analyses mentioned across the manuscript have been deposited on figshare: https://doi.org/10.6084/m9.figshare.28801289. Supplementary File 1 - Supplementary File 36 are available at https://doi.org/10.6084/m9.figshare.28801289.

The following dataset was generated:

| Author(s) | Year | Dataset title | Dataset URL | Database and Identifier |
|---|---|---|---|---|
| Garcia De Las Bayonas A | 2025 | Supplementary files - G protein-coupled receptor diversity and evolution in the closest living relatives of Metazoa | https://doi.org/10.6084/m9.figshare.28801289 | figshare, 10.6084/m9.figshare.28801289 |

The following previously published datasets were used:

| Author(s) | Year | Dataset title | Dataset URL | Database and Identifier |
|---|---|---|---|---|
| Richter DJ, Fozouni P, Eisen MB, King N | 2018 | Data from: Gene family innovation, conservation and loss on the animal stem lineage | https://doi.org/10.6084/m9.figshare.5686984.v2 | figshare, 10.6084/m9.figshare.5686984.v2 |
| Brunet T, Larson BT, Linden TA, Vermeij MJA, McDonald K, King N | 2019 | Choanoeca flexa transcriptome and predicted nonredundant proteome | https://doi.org/10.6084/m9.figshare.8216291.v2 | figshare, 10.6084/m9.figshare.8216291.v2 |
| Sebé-Pedrós A, Irimia M, del Campo J, Parra-Acero H, Russ C, Nusbaum C, Blencowe BJ, Ruiz-Trillo I | 2013 | Genome - Capsaspora owczarzaki (v3) | https://doi.org/10.6084/m9.figshare.4123158.v2 | figshare, 10.6084/m9.figshare.4123158.v2 |
| Ocaña-Pallarès E, Multicellgenome Lab | 2022 | Genomic data for Ministeria vibrans, Parvularia atlantis, Pigoraptor vietnamica and Pigoraptor chileana | https://doi.org/10.6084/m9.figshare.19895962.v1 | figshare, 10.6084/m9.figshare.19895962.v1 |
| Department of Life and Environmental Sciences, Prefectural University of Hiroshima | 2019 | Sphaeroforma arctica strain 3-2015 genome | https://www.ncbi.nlm.nih.gov/bioproject/563213 | NCBI BioProject, 563213 |
| de Mendoza A, Suga H, Permanyer J, Irimia M, Ruiz-Trillo I | 2015 | Creolimax fragrantissima genome data | https://doi.org/10.6084/m9.figshare.1403592 | figshare, 10.6084/m9.figshare.1403592 |
| Grau-Bové X, Torruella G, Donachie S, Suga H, Leonard G, Richards TA, Ruiz-Trillo I | 2017 | Genome - Ichthyophonus hoferi | https://doi.org/10.6084/m9.figshare.5426488 | figshare, 10.6084/m9.figshare.5426488 |

*Continued*

| Author(s) | Year | Dataset title | Dataset URL | Database and Identifier |
|---|---|---|---|---|
| Grau-Bové X, Torruella G, Donachie S, Suga H, Leonard G, Richards TA, Ruiz-Trillo I | 2017 | Genome - Abeoforma whisleri | https://doi.org/10.6084/m9.figshare.5426458 | figshare, 10.6084/m9.figshare.5426458 |
| Grau-Bové X, Torruella G, Donachie S, Suga H, Leonard G, Richards TA, Ruiz-Trillo I | 2018 | Genome - Pirum gemmata | https://doi.org/10.6084/m9.figshare.5426506 | figshare, 10.6084/m9.figshare.5426506 |
| Grau-Bové X, Torruella G, Donachie S, Suga H, Leonard G, Richards TA, Ruiz-Trillo I | 2017 | Genome - Chromosphaera perkinsii | https://doi.org/10.6084/m9.figshare.5426494 | figshare, 10.6084/m9.figshare.5426494 |
| Hehenberger E, Tikhonenkov DV, Kolisko M, del Campo J, Esaulov AS, Mylnikov AP, Keeling PJ | 2018 | Data from: Novel predators reshape holozoan phylogeny and reveal the presence of a two-component signalling system in the ancestor of animals | https://doi.org/10.5061/dryad.26bv4 | Dryad Digital Repository, 10.5061/dryad.26bv4 |

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

## Appendix 1

### Conservation of the GPCR signaling pathway in choanoflagellates

In Metazoa, the binding of a ligand to the N-terminus or extracellular loops of a G-protein-coupled receptor (GPCR) triggers a conformational change that initiates the rest of the GPCR signaling pathway (*Hilger et al., 2018*; *Wootten et al., 2018*; *Hauser et al., 2021*). Ligand-bound GPCRs act as guanine nucleotide exchange factors (GEFs) for the GDP-bound Gα subunit, catalyzing the exchange of GDP for GTP and, thereby, promoting the dissociation of activated Gα-GTP from Gβγ dimers (Figure S3A, *Hilger et al., 2018*; *Weis and Kobilka, 2018*; *Wootten et al., 2018*). Both activated Gα-GTP and free Gβγ dimers can transduce the signal independently to downstream effectors (*Marinissen and Gutkind, 2001*; *Neves et al., 2002*; *Dupré et al., 2009*). Ric8 proteins and Phosducin facilitate the folding and targeting at the membrane of Gα subunit and Gβγ dimers, respectively (*Willardson and Howlett, 2007*; *Srivastava et al., 2019*). Overstimulation of GPCR signaling is detrimental to the cell, and various mechanisms quickly turn off GPCR activity in Metazoa. Signal termination is controlled in part by the hydrolysis of Gα-GTP to Gα-GDP, which reconstitutes the inactive Gαβγ heterotrimer. The rate of Gα-GTP hydrolysis is a consequence of the intrinsic GTPase activity of Gα and can be enhanced by the action of regulators of G protein signaling (RGS), proteins that act to accelerate the GTPase activity (*O'Brien et al., 2019*). In addition, GoLoco motif proteins are negative regulators of Gα-GTP signaling, as they slow the spontaneous release of GDP from the Gα subunit (*Willard et al., 2004*; *Sato et al., 2006*). Another major axis to terminate GPCR signaling involves the rapid phosphorylation of the ligand-bound receptor by GPCR-Kinases (GRKs) and the subsequent binding of Arrestin proteins that selectively recognize active phosphorylated receptors (*Rajagopal and Shenoy, 2018*; *Gurevich and Gurevich, 2019*). Arrestins not only abolish G-protein-mediated signaling but also act as a scaffold to facilitate multiple downstream signaling pathways and trigger the endocytosis of the receptor (*Figure 1—figure supplement 3A*, *Bagnato and Rosanò, 2019*).

We found that the heterotrimeric G proteins Gα, Gβ, and Gγ, along with positive regulators (Ric8, Phosducin) and negative regulators (GRK, Arrestin, RGS, GoLoco) of GPCR signaling, were detected in most choanoflagellates, with some exceptions in a few species (*Figure 1—figure supplement 3B*, Supplementary file 5). Notably, we failed to detect Gγ and GRK in nearly half of the choanoflagellates, which could either suggest real losses, false negatives, or a divergence beyond recognition of these genes in the species concerned.

We decided to assess further the diversity of Gα subunits, the best-characterized transducers of metazoan GPCR signaling, in choanoflagellates. In metazoans, Gα subunits are classified into five main families (Gαs, Gαi/o, Gαq/11, Gα12/13, and Gαv), and members of each family interact with different effectors to produce distinct cellular responses (*Figure 1—figure supplement 3C*, *Marinissen and Gutkind, 2001*; *Neves et al., 2002*; *Oka et al., 2009*; *Oka and Korsching, 2009*; *Doktorgrades and Obaid, 2022*; *Zhang et al., 2024*). Although all major G protein Gα classes (Gα$_s$, Gα$_{q/11}$, Gα$_{12/13}$, Gα$_v$, and Gα$_{i/o}$) likely originated prior to the diversification of metazoans from the rest of holozoans (*de Mendoza et al., 2014*; *Krishnan, 2015*; *Lokits et al., 2018*), our phylogenetic analysis recovered only three Gα classes in choanoflagellates (Gα$_s$, Gα$_{q/11}$, and Gα$_v$; *Figure 1—figure supplement 3D* and Supplementary files 6 and 7). Interestingly, Gα$_s$ subunits, previously thought to be lost in choanoflagellates, were detected in four closely related species (*Salpingoeca kvevrii*, *Salpingoeca urceolata*, *Salpingoeca macrocollata*, and *Salpingoeca punica*). Based on their functions in metazoans, the activation of these Gα subunits could modulate protein phosphorylation (Gαs and Gαq/11), calcium signaling (Gαq/11), and ion homeostasis (Gαv) downstream of the GPCRs in choanoflagellates (*Figure 1—figure supplement 3C and D*; *Marinissen and Gutkind, 2001*; *Neves et al., 2002*; *Abu Obaid et al., 2024*).

Overall, our results suggest that the GPCR signaling pathway is conserved in most choanoflagellate species, providing additional evidence that metazoan-like G protein signaling is part of the repertoire of signaling activities in choanoflagellates (*de Mendoza et al., 2014*; *Krishnan et al., 2015*; *Lokits et al., 2018*).

