## [Editor Report · eLife Assessment]

This **important** study fills a gap in our knowledge of the evolution of GPCRs in holozoans, as well as the phylogeny of associated signaling pathway components such as G proteins, GRKs, and RIC8 proteins. The evidence supporting the conclusions is **compelling**, with the analysis of extensive new genomic data from choanoflagellates and other non-animal holozoans. Overall, the study is thorough and well-executed. It will be a resource for researchers interested in both the comparative genomics of multicellularity and GPCR biology more broadly, especially given the importance of GPCRs as highly druggable targets

---

## [Referee Report · Reviewer #1 (Public review)]

Summary:

The authors strived for an inventory of GPCRs and GPCR pathway component genes within the genomes of 23 choanoflagellates and other close relatives of metazoans.

Strengths:

The authors generated a solid phylogenetic overview of the GPCR superfamily in these species. Intriguingly, they discover novel GPCR families, novel assortments of domain combinations, novel insights into the evolution of those groups within the Opisthokonta clade. A particular focus is laid on adhesion GPCRs, for which the authors discover many hitherto unknown subfamilies based on Hidden Markov Models of the 7TM domain sequences, which were also reflected by combinations of extracellular domains of the homologs. In addition, the authors provide bioinformatic evidence that aGPCRs of choanoflagellates also contained a GAIN domain, which are self-cleavable thereby reflecting the most remarkable biochemical feat of aGPCRs.

Weaknesses:

The chosen classification scheme for aGPCRs may require reassessment and amendment by the authors in order to prevent confusion with previously issued classification attempts of this family.

---

## [Referee Report · Reviewer #2 (Public review)]

Summary:

The authors set out to characterise the GPCR family in choanoflagellates (and other unicellular holozoans). GPCRs are the most abundant gene family in many animal genomes, playing crucial roles in a wide range of physiological processes. Although they are known to evolve rapidly, GPCRs are an ancient feature of eukaryotic biology. Identifying conserved elements across the animal-protist boundary is therefore a valuable goal, and the increasing availability of genomes from non-animal holozoans provides new opportunities to explore evolutionary patterns that were previously obscured by limited taxon sampling. This study presents a comprehensive re-examination of GPCRs in choanoflagellates, uncovering examples of differential gene retention and revealing the dynamic nature of the GPCR repertoire in this group. As GPCRs are typically involved in environmental sensing, understanding how these systems evolved may shed light on how our unicellular ancestors adapted their signalling networks in the transition to complex multicellularity.

Strengths:

The paper combines a broad taxonomic scope with the use of both established and recently developed tools (e.g. Foldseek, AlphaFold), enabling a deep and systematic exploration of GPCR diversity. Each family is carefully described, and the manuscript also functions as an up-to-date review of GPCR classification and evolution. Although similar attempts of understanding GPCR evolution were done over the last decade, the authors build on this foundation by identifying new families and applying improved computational methods to better predict structure and function. Notably, the presence of Rhodopsin-like GPCRs in some choanoflagellates and ichthyosporeans is intriguing, even though they do not fall within known animal subfamilies. The computational framework presented here is broadly applicable, offering a blueprint for surveying GPCR diversity in other non-model eukaryotes (and even in animal lineages), potentially revealing novel families relevant to drug discovery or helping revise our understanding of GPCR evolution beyond model systems.

Weaknesses:

While the study contributes several interesting observations, it does not radically revise the evolutionary history of the GPCR family. However, in an era increasingly concerned with the reproducibility of scientific findings, this is arguably a strength rather than a weakness. It is encouraging to see that previously established patterns largely hold, and that with expanded sampling and improved methods, new insights can be gained-especially at the level of specific GPCR subfamilies. Then, no functional follow ups are provided in the model system *Salpingoeca rosetta*, but I am sure functional work on GPCRs in choanoflagellates is set to reveal very interesting molecular adaptations in the future.

Comments on the latest version:

The authors have done a good job answering my questions and suggestions.

---

## [Author Response]

The following is the authors’ response to the previous reviews

**Reviewer #1:**
“I am sorry to dwell on the point of naming the newly identified families of adhesion GPCRs in choanoflagellates. I commented: "Can the authors suggest another scheme (mind to avoid the subfamily I-IX or the alternative ADGRA-G,L,V subfamily schemes of metazoan aGPCRs) and adapt their numbering throughout the text and all figures/supplementary figures/supplementary files." Now the authors have changed the Roman numeral numbering (previously used by the adhesion GPCR field to denominate metazoan receptor families) to the other option that I explicitly said should be obsolete, the numbering by capital letters (which is in use since its introduction in 2015 in Hamann et al., Pharmacol Rev, 2015). The authors write: "Phylogenetic analysis of the 7TM domains of choanoflagellates uncovered at least 19 subfamilies of aGPCRs subfamilies A-S ...". I am thus afraid this has not addressed my point at all. For example, in the revised numbering scheme for Choanoflagellates aGPCR subfamilies of the authors the now used "A" descriptor, which are predicted to contain a HYR domain, can be mistaken for ADGRA homologs (abbreviated as "A" receptors, previously termed subfamily III aGPCRs) of metazoan aGPCRs, which contain HRM and LRR domains. Likewise, choanoflagellate "E" receptors are predicted to harbour LRR repeats, but metazoan ADGRE (abbreviated as "E" too) are characterised by their EGF domains. This clearly underlines the need to devise a numbering scheme for the newly described choanoflagellate aGPCR homologs so they cannot be confused with the receptors from other kingdoms, for which identical naming conventions exist. Please change this, e.g. by numbering/denominating the choanoflagellate subfamilies by greek letters (or your pick of any other ordering system that does not lend itself to be mistaken with the previous and existing aGPCR classifications) and change the manuscript and figures accordingly.”

We have now re-labeled the choanoflagellate aGPCR subfamilies, previously numbered from A to S, using Greek alphabetical enumeration (from α to τ). Changes have been made throughout the main text, in Figure 5, and in Supplementary Figures S6 and S7.